

Synoptic Ozone, Cloud Reflectivity, and Erythemal Irradiance from Sunrise to Sunset for the Whole Earth
2         as viewed by the DSCOVR spacecraft from Lagrange-1

Jay Herman[1], Liang Huang[2], Richard McPeters[3], Jerry Ziemke[3], Alexander Cede[4], Karin Blank[3]
Abstract
The EPIC instrument onboard the DSCOVR spacecraft, located near the Earth-Sun gravitational plus
centrifugal force balance point, Lagrange-1, measures Earth reflected radiances in 10 wavelength
channels ranging from 317.5 nm to 779.5 nm. Of these channels, four are in the UV range 317.5, 325,
340, and 388 nm, which are used to retrieve $O_3$, 388 nm scene reflectivity (LER Lambert Equivalent
Reflectivity), $SO_2$, and aerosol properties. These quantities are derived synoptically for the entire sunlit
globe from sunrise to sunset every 68 minutes or 110 minutes for summer or winter at the receiving
antenna in Wallops Island, Virginia, respectively. Depending on solar zenith angle, either 317.5 or 325
nm channels are combined with 340 and 388 nm to derive ozone amounts. As part of the ozone
algorithm, the 388 nm channel is used to derive LER.   The retrieved ozone amounts and LER are
combined to derive the erythemal irradiance for the sunlit Earth's surface at a resolution of 18 x 18 $km^2$
near the center of the Earth's disk using a computationally efficient approximation to a radiative transfer
calculation of irradiance.   Corrections are made for altitude above sea level and for the reduced
transmission by clouds based on retrieved LER.
[1]University of Maryland Baltimore County, Maryland
[2]Science Systems and Applications, Lanham, Maryland
[3]NASA Goddard Space Flight Center, Greenbelt, Maryland
[4]SciGlob Instruments and Services, Maryland



DSCOVR/EPIC Synoptic Ozone, Cloud Reflectivity, and Erythemal Irradiance From Sunrise to Sunset for
the Whole Earth as viewed from an Earth-Sun Lagrange-1 Orbit

## 1.0 Introduction

The DSCOVR (Deep Space Climate Observatory) spacecraft was successfully launched on 11
February 2015 to an orbit near the Earth-Sun gravitational plus centrifugal force balance point,
Lagrange-1 (L-1), $1.5 \times 10^6$ km from the Earth. The earth pointing instruments on the DSCOVR spacecraft
placed in orbit about the L-1 point will simultaneously observe the sun illuminated earth's disk from
sunrise to sunset. An illustration of the orbit is given in the Appendix  (see https://epic.gsfc.nasa.gov for
details). DSCOVR started to transmit Earth data after it achieved a quasi-stable orbit in mid-June 2015.
The DSCOVR mission at L-1 is optimum for early warning solar flare observations (magnetic field,
electron, and proton fluxes) from instruments contained on the sunward side of DSCOVR, and contains
two Earth-viewing instruments allowing continuous observation of the sunlit face of the Earth.  The EPIC
(Earth Polychromatic Imaging Camera) instrument onboard DSCOVR images the Earth in ten narrow
band wavelength channels (up to 2048 x 2048 pixels), producing both color images of the Earth and
science data products such as ozone, $SO_2$, aerosol amounts, cloud reflectivity, UV surface irradiance,
cloud and aerosol heights, and vegetation indices. This paper discusses the UV science products $O_3$,
cloud reflectivity, and UV surface irradiance, methods of retrieval, and EPIC's UV in-flight calibration.

### 1.1 EPIC Instrument

The EPIC instrument consists of a 30-cm aperture 283.642 cm focal length Cassegrain telescope
containing a multi-element field-lens group focusing light onto a UV sensitive 2048 x 2048 hafnium
coated CCD detector with 12 bit readout electronics. Images are made through ten narrow-band filters,
four in the ultraviolet, four in the visible, and two in the near infrared. The 10 filter transmission
functions are shown in Fig. 1.  Observations are made as light passes sequentially through each of ten
narrow-band filters mounted in two moveable filter wheels and through an exposure control 3-slot
rotating shutter. The exposure times for each wavelength were adjusted in-flight to achieve an
approximately 80 % CCD electron well fill in the brightest scenes, which were observed during the first
week of operation, to avoid saturation and leaking from one pixel to another (blooming). Earth exposure
times range from about 654 milliseconds at 317.5 nm to 22 milliseconds at 551 nm, which have not
changed during the current life of the mission. Another set of exposure times was determined for
viewing the full moon as seen from the Earth (Table 1). The CCD has a well depth of approximately
$8.5 \times 10^4$ electrons (a maximum signal to noise ratio SNR of 290:1) before a small dark current correction
that is a function of its in-flight operating temperature of $-20^O$C.  The 12-bit readout means that there
are $2^{11}$ (2048) readout steps or counts (42 electrons/count). The counts divided by the exposure time
(counts/second) are converted to radiances or albedos using in-flight scene matching calibration from
low earth orbit satellites (see Sect. 1.2 and Table 2). The maximum SNR applies to the brightest of
scenes over high clouds or fresh snow over ice. Cloud-free and snow-free scenes have much lower SNR,
which affects the visible channels more than the UV channels because of the lower scene contrasts with
clouds caused by enhanced UV Rayleigh scattering. There are occasional bright flashes caused by ice



crystals in high clouds that saturate a few pixels (see Fig. 2 and Marshak et al., 2017) in the equatorial
and mid-latitude regions.
The filters of interest for calculating ozone amounts, aerosol index, and cloud reflectivity are
centered on 317.5, 325, 340, and 388 nm in the wavelength band with full widths at half maximum
(FWHMs) 1.0, 1.0, 2.7, and 2.6 nm, respectively.  For the UV channels, 2 x 2 individual pixels are
averaged onboard the spacecraft to yield an effective 1024 x 1024 pixel image corresponding to an 18 x
18 km$^2$ resolution at the observed center of the Earth's sunlit disk. The effective spatial resolution
decreases as the secant of the angle between EPIC's sub-earth point and the normal to the earth's
surface. Only the 443 nm channel is retrieved at full resolution to help with resolving cloud cover and
obtaining improved color images. The sampling resolution of a single pixel is about 8 x 8 km$^2$ (about 1
arcsecond), but including the effect of the optical point-spread function, the effective 443 nm channel
resolution is about 10 km. The effective resolution at 443 nm has been verified by looking at clear
scenes over the Nile River in Egypt and, occasionally, the cloud-free Amazon River in Brazil.
EPIC data has been obtained since June 15, 2015 at a rate of one set of 10 wavelengths every 68
minutes during Northern Hemisphere (NH) summer and one set every 110 minutes in the winter. The
difference between summer and winter rates is caused by the reduced number of hours in the winter
when the antenna (located at Wallops Island, Virginia) is in view of the spacecraft, and limitations from
the spacecraft memory technology from the late 1990s.
Each of the 10-wavelength measurements is obtained at a slightly different times. The first filter
in the sequence is 443 nm, which takes about 2 minutes to complete a measurement (28 ms exposure
time (Table 1) plus CCD readout and onboard processing time that includes 12-bit jpeg compression of a
2048 x 2048 pixel image). The remaining 9 filter measurements take a total of about 5 minutes
(exposure times plus CCD readout into memory) and then another 13 minutes to process the data for
the 9 filters (this includes 12-bit jpeg compression of 1024 x 1024 images that have been averaged
onboard in groups of 2x2 pixels before compression). Adjacent pairs of wavelengths are measured at 30
second intervals before the onboard processing is started. This means the individual channel images are
not co-located at the pixel level because of earth rotation (15.03$^{\circ}$ per hour or about 1670 km per hour
at the equator), the slow rotation of the spacecraft, 0.082$^{\circ}$ per hour, and a small amount of spacecraft
jitter). Each pixel views about 1 arc second or 2.78x10$^{-4}$ degrees. Data from an onboard star-tracker and
feedback from the earth's image on the CCD keep the images approximately centered on the CCD. The
lack of native channel-to-channel colocation requires an elaborate spherical geometry geolocation
analysis to adjust the data to a common latitude x longitude grid with an accuracy of 1/4 of a pixel.
A description of the EPIC instrument, its orbit, and some of the data products can be obtained
from http://avdc.gsfc.nasa.gov/pub/DSCOVR/Web_EPIC/ and from http://epic.gsfc.nasa.gov/. The EPIC
raw  counts/second  and  science  data  (Version  2  used  in  this  paper)  are  archived  at
https://eosweb.larc.nasa.gov/project/dscovr/dscovr_table in HDF5 format.



This paper presents examples of the ozone and scene reflectivity retrievals that are used to
obtain unique estimates of erythemal UV irradiance (or UV Index, UVI) as a function of latitude,
longitude, local solar time (LST), and altitude above sea level (ASL). Since this is the first paper on EPIC
retrieved ozone, Sect. 1 contains a brief description of the calibration of the four UV channels and the
ozone retrieval algorithm. Sect. 2 shows examples of natural color images, Sect. 3 gives an example of
retrieved ozone and the corresponding 388 nm Lambert Equivalent Reflectivity (LER, Herman et al.,
2009), Sect. 4 presents a validation of EPIC retrieved ozone compared to ozone from ground-based and
satellite data, Sect. 5 shows details of the latitudinal and longitudinal synoptic variability of ozone, and
Sect. 6 presents new results showing the sunrise to sunset variability of UV erythemal radiation reaching
the Earth's surface including the reduction by clouds from sunrise to sunset.
The data and images of the changing synoptic cloud cover from sunrise to sunset are unique to
the EPIC satellite instrument.  Neither geostationary nor low earth orbiting satellites can produce these
data or images.  Geostationary satellites could produce something similar, but to date, none have the UV
channels for ozone and LER, and geostationary satellites are limited to a range of approximately $\pm 60^O$
latitude and $\pm 60^O$ longitude.  While low earth orbiting satellite data can be combined to produce a global
representation of ozone and cloud cover, all the ozone and cloud cover are for a fixed local time (e.g.,
13:30 hours for OMI) and is not representative of the atmosphere at other times of the day.
**1.2 Calibration**
Before the raw EPIC data (counts per second) can be used, a number of pre-processing steps
must be accomplished. The major steps are 1) measuring and subtracting the dark current signal, 2)
"flat-fielding" the CCD so that the sensitivity differences between all four million pixels are determined
and corrected, 3) correcting for stray-light effects to account for light that should be going to a particular
pixel, but instead is scattered to different pixels, and 4) determining the radiometric calibration for each
wavelength channel in terms of EPIC counts/second to be converted to earth normalized radiances or
reflectances (backscattered at approximately $172^O$). The earth upwelling normalized radiance $I_M$ (W/(m$^2$
nm sr)) at the top of the atmosphere (TOA) is defined in terms of the albedo $A_M$ given by Eq. 1,

$$A_M = \frac{I_M}{S_M / D_E^2} \qquad (\text{sr}^{-1}) \qquad\qquad (1)$$

for wavelength bands M=1 to 4, $S_M$ is the incident solar irradiance (W/(m$^2$ nm)) weighted with the filter
function for band M at 1 AU and $D_E$ is the sun-earth distance in AU (astronomical units).  Since EPIC does
not measure solar irradiance, we use a high resolution solar irradiance spectrum, $S(\lambda)$ (Dobber et al.,
2008), as a reference solar spectrum.  The reference spectrum is weighted with EPIC's filter transmission
functions $T_M(\lambda)$  (Fig. 1) to obtain each EPIC channel's weighted solar irradiance $S_M$ at solar-earth
distance at 1 astronomical unit (Eqs. 1 and 2).

$$S_M = \int_{\lambda_1}^{\lambda_2} T_M(\lambda) S(\lambda) d\lambda \Big/ \int_{\lambda_1}^{\lambda_2} T_M(\lambda) d\lambda \qquad (\text{Wm}^{-2}\text{nm}^{-1}) \qquad (2)$$




137   In-flight radiometric calibration is accomplished by comparison with albedo values measured by
138 current well-calibrated LEO (low-earth orbiting; e.g., Aura/OMI, Ozone Monitoring Instrument, and
139 Suomi-NPP/OMPS, National Polar-orbiting Partnership/Ozone Mapping and Profiler Suite) satellite
140 instruments observing scenes that match in time and observing angles with those from EPIC. For albedo
141 measurements, OMPS has a calibration accuracy of 2 %, while its wavelength dependence (precision) in
142 the calibration is estimated to be better than 1 % (Jaross et al., 2014). The OMPS Nadir Mapper on
143 Suomi-NPP has a 50 x 50 $km^2$ footprint in its normal operating mode with 36 cross-track views ($\pm55^o$
144 satellite view angle or strip of about $\pm12^o$ equatorial longitude). It has a spectral resolution of 1 nm,
145 which is close to EPIC's 317.5 nm and 325 nm channels FWHM, but narrower than EPIC's 340 nm and
146 388 nm channels. To perform in-flight calibration, OMPS' albedo spectra were either interpolated (for
147 317.5 and 325 nm channels) or convolved (at 340 and 388 nm) with each EPIC filter transmission
148 function $T_M$ (Fig. 1). Because the albedo spectra $A_M(\lambda)$ (Eq. 1) cancels the solar irradiance $S_M$ Fraunhofer
149 line structure, the interpolation and convolution of $A_M(\lambda)$ has better accuracy than directly using the
150 radiance spectra $I_M(\lambda)$. OMI on Aura has 13 x 24 $km^2$ spatial resolution and about $\pm56^o$ cross-track views
151 (a strip of ± 1300 km or $\pm 13^o$ equatorial longitude) with a spectral resolution of 0.42 nm. To match
152 measurements with DSCOVR, OMI's albedo spectra were convolved with EPIC's $T_M(\lambda)$. Then, the results
153 in every two adjacent cross-track views and four consecutive along-track scans are combined to form 50
154 x 50 $km^2$ footprints for comparison with EPIC measured counts/second obtained from 7 x 7 EPIC pixels.

155   EPIC raw counts/second inside each coincident footprint are preprocessed by the steps stated in
156 a previous paragraph. Then, the counts/second average and variance in each coincident footprint are
157 computed to obtain the EPIC albedo calibration coefficients $K_M$ (Eq. 3). Misalignment between EPIC and
158 OMPS or OMI footprints can result large scene noise unless uniform scenes are selected and less
159 uniform scenes discarded. This is achieved by weighting each coincident data point with the reciprocal
160 of the percent EPIC counts/second variance inside the coincident footprint. All of the coincident points
161 between LEO satellites and EPIC observations occur within $\pm40^o$ of the earth's equator. Selected LEO
162 footprints have viewing angles nearly identical to EPIC's (within $1^o$ in backscatter angle and $2^o$ degrees in
163 solar zenith angle). EPIC's backscatter angle varies with latitude and longitude by less than $0.25^o$, since
164 the angular size of the earth varies from $0.45^o$ to $0.53^o$ to $0.45^o$ every 6 months depending on the
165 location of DSCOVR in its orbit (an irregular Lissajous orbit about L-1 that is tilted relative to the ecliptic
166 plane and perturbed by the Earth's moon). The orbit varies from $4^o$ to $15^o$ away from the Earth-Sun line.
167 These small differences in observing geometry are corrected in the atmospheric radiative transfer model
168 calculations $\alpha(\lambda)$ (Eq. 4), resulting in corrections less than 2 %. EPIC albedo calibration coefficients are
169 derived from Eqs. 3 and 4.

$$K_M = \frac{A_M(OMPS)\{\alpha_M(EPIC)/\alpha_M(OMPS)\}}{C_M(EPIC)D_E^2} \qquad (3)$$

$$\alpha_M = \int \alpha(\lambda)S(\lambda)T_M(\lambda)\,d\lambda \Big/ \int S(\lambda)T_M(\lambda)\,d\lambda \qquad (4)$$


171 where





M is the EPIC channel number, M=1,2,3,4
$A_M(OMPS)$ = OMPS albedo measurement in the EPIC channel-M wavelength band
$\alpha_M(EPIC)$ and $\alpha_M(OMPS)$ are computed albedo values for EPIC and OMPS coincident geometry,
$C_M(EPIC)$ is the average count rate over the pixels matching OMPS,
$D_E$ is the sun-earth distance in AU.
$\alpha(\lambda)$ is the computed high resolution normalized radiance spectrum,
$S(\lambda)$ is the referenced high resolution solar irradiance spectrum,
$T_M(\lambda)$ is the EPIC filter transmission profile or the OMPS slit function.
All of the coincidence points with LEO satellite instruments were measured using the area of
the EPIC CCD within 600 pixels of its center. There are about 15000 coincidence data points accumulated
by the end of 2016.  Because of the large number of data points, statistical averaging errors are small.
An atmospheric radiative transfer model, RTM, takes total column ozone and surface reflectivity from
LEO retrievals to obtain both $\alpha_M(EPIC)$ and $\alpha_M(LEO )$.  Although uncertainties in the RTM can propagate
into the computed albedos, the resulting uncertainties in $\alpha_M(EPIC)$ and $\alpha_M(LEO)$ are approximately
identical, and approximately cancel in Eq. 3.  The resulting EPIC albedo calibration uncertainty is mostly
inherited from the OMPS albedo calibration uncertainty, which has an accuracy of 2 % and a precision of
1 % in relative (wavelength dependent) values.  For the UV channels, the calibration factors $K_M$ are not
constants, but are slowly increasing functions of time (on average 0.016 per year; see $K_M(t)$ in Fig. 2),
which is normalized to one on 1 January 2016).  Table 2 shows the reference values of $K_M$ multiplied by
$\pi$.
Using Tables 1, 2, and Fig. 2, EPIC albedo measurements are derived with

$$A_M(EPIC) = K_M C_M(EPIC) D_E^2 \qquad\qquad (1\text{-}5)$$


Note that the factor $D_E^2$ for solar irradiance at 1 AU is contained in the albedo calibration
coefficient $K_M$.  Since solar activity changes (e.g., 27.5 day cycle) are negligible for EPIC UV channel
wavelengths, daily solar irradiance changes are only adjusted with the sun-earth distance $D_E$.  Users of
EPIC data may also be interested in radiance measurements.  The radiance calibration coefficients can
be derived with Eq. 6,

$$E_M = K_M S_M \qquad\qquad (6)$$

and the radiance measurements can be obtained with Eq. 7.

$$I_M(EPIC) = E_M C_M(EPIC) \qquad\qquad (7)$$

The uncertainty in the radiance calibration can increase significantly due to errors in estimating
the absolute solar irradiance.  Uncertainty in estimated $S_M$ for EPIC UV channels in Table 1 is about 3 %.



### 1.3 Ozone Algorithm

Once the albedo calibration factors are applied to EPIC's measured counts/second, the calculated albedos can be combined to retrieve total column ozone (TCO), Lambert Equivalent Reflectivity (LER), and aerosol index (AI). The TOA directional albedo calculation uses the TOMRAD radiative transfer calculation code, which has a spherical geometry correction for large solar zenith angles (SZA) and satellite looking angles (SLA) (Caudill et al., 1997). The calculation uses the same climatological ozone profiles used in OMI retrievals, altitude weighted average effective ozone temperatures, ground reflectivities, terrain height, and climatological cloud heights. Spectrally resolved $O_3$ absorption cross sections are from Brion et al., (1993, 1998); Daumont et al., (1992); and Malicet et al., (1995). The resulting spectra are convolved with the EPIC filter transmission functions (Fig. 1) and with the reference solar irradiance spectra (see Eq. 4).

The resulting computed $\alpha_M$ (Eq. 4) are compiled into a finely stepped look-up table as functions of ozone profiles and solar-view angles. EPIC ozone retrieval uses the 388 nm channel for computing the surface reflectivity with a formula similar (except for choice of wavelengths) to that used in cloud reflectivity studies (Herman et al., 2009). Then, the retrieval is based on two ozone absorption channels, 317.5 nm and 340 nm for low optical depth conditions, or 325 nm and 340 nm for high optical depth conditions, together with the 388 nm measurement to form triplet equations. The ozone retrieval algorithm assumes a linear wavelength dependence in the surface reflectivity (Eq. 8),

$$R_\lambda = R_{\lambda_0} + b(\lambda - \lambda_0) \tag{8}$$

where $\lambda_0$ is given wavelength 388 nm. The total column ozone (TCO) is given by Eq. 9,

$$\Omega = \Omega_0 + \frac{\Delta N_{\lambda_1} \frac{\partial N_{\lambda_2}}{\partial R}(\lambda_2 - \lambda_0) - \Delta N_{\lambda_2} \frac{\partial N_{\lambda_1}}{\partial R}(\lambda_1 - \lambda_0)}{\frac{\partial N_{\lambda_1}}{\partial \Omega}\frac{\partial N_{\lambda_2}}{\partial R}(\lambda_2 - \lambda_0) - \frac{\partial N_{\lambda_2}}{\partial \Omega}\frac{\partial N_{\lambda_1}}{\partial R}(\lambda_1 - \lambda_0)} \tag{9}$$

where

$\Omega_0$ is an initial climatology estimate of TCO or TCO from previous step in the iteration,

$\lambda_1$ and $\lambda_2$ are the selected ozone absorption wavelengths,

$N_\lambda$ is the N-value defined as logarithm of the albedo values by Eq. 10,

$$N_\lambda = -100\, log_{10}\{I_\lambda/(S_\lambda/D^2)\} \tag{10}$$

and

$\Delta N_\lambda$ is the N-value residue (difference between the measured N-value and the computed N-value),



$\dfrac{\partial N_{\lambda x}}{\partial Z}$ = measurement sensitivity with respect to the total column ozone, Z = Ω, or the surface
reflectivity, Z = R, for wavelengths $\lambda_1$ or $\lambda_2$.

If one assumes the sensitivities to the surface reflectivity, $\partial N_\lambda / \partial R$ are wavelength independent,

Eq. 5 for the triplet algorithm is similar to the Version 8 TOMS algorithm (Rodriguez et al., 2003).

Since the algorithm for ozone (Eqs. 8 to 10) requires the use of two or more wavelength

channels, the measured counts/second for each channel must be geolocated on a common latitude x
longitude grid that is accurate to 0.25 of a single pixel size. When projected on the 3-D Earth, the
sampling size is about 8 km at nadir and effectively increases to 10 km when EPIC's point spread
function is applied.  The result for 2 x 2 pixel averaging is a spatial resolution at nadir of about 18 km,
which gets larger as the secant of the SLA from the nadir point. SLA is measured relative to the normal
to the Earth's surface, and is $0^O$ at nadir and almost $90^O$ at the Earth's sunlit terminator.  The radiative
transfer spherical geometry correction is accurate to about $80^O$ in SZA and SLA, which means that
retrieved ozone values near the Earth's terminator are not accurate.

**2    Natural Color Images**

A typical eye response color image view of the Earth, obtained by a weighted combination of

the geolocated red, green, and blue wavelength channels, is shown in  Fig. 2. To produce RGB images
adjusted to the human eye response, the algorithm used is a derivative of the International Commission
on Illumination (CIE) process for estimating tristimulous values from calibrated instruments (Wyszecki
and Stiles, 1982; Broadbent, 2004; Gardner, 2007; Bodrogi and Khanh, 2012). Obtaining eye response
images for EPIC's narrow band filters (Table 1) was improved by customization of the algorithm to use
additional channels than just the 443, 551, and 680 nm blue, green, and red channels.

Because the blue 443 nm channel is not spatially averaged onboard the spacecraft, the color

images have a maximum resolution of about 10 km at nadir determined by looking at the discernable
width of the Nile and Amazon Rivers. The color images also give an indication of the quality of the
geolocation. Errors in geolocation would appear as pink edges at the cloud boundaries, which are not
present in the images in Figs. 3 or in the complete image collection on http://epic.gsfc.nasa.gov/.

Even with accurate geolocation, about 0.25 pixels (2 km), between the 4 UV channels, there is

some noise introduced into ozone retrievals by small cloud edge location errors when transferring all of
the native data to a common latitude and longitude grid. Ozone retrievals over almost cloud-free
scenes, such as over the Saharan desert or clear-sky portions of the oceans, show much less noise than
those with partial cloud cover. Since the pixel-to-pixel noise caused by misaligned cloud edges is almost
random, spatial averaging to about 50x50 $km^2$ (similar to TOMS and OMPS, but coarser than OMI spatial
resolution) reduces the effect of apparent noise from cloud edges.  The following sections use 25 x 25
$km^2$ spatial averaging (3 x 3 CCD pixels), which has more spatial details and some cloud-edge noise
(noise < 3 %).



### 3  Examples of EPIC Ozone and Reflectivity


A matched pair of images for ozone and scene reflectivity LER (17 April 2016) are shown in Fig. 4
with a maximum resolution of 18 km, since all UV channels involved in the ozone retrieval are
downlinked from the spacecraft at a resolution of 2 x 2 onboard averaged pixels. Note that the reduced
resolution hdf5 data files stored on the ground are in their original sampling density (2048 x 2048), but
have reduced spatial resolution. In Fig. 4, the entire data image for ozone and the LER scene reflectivity
are all at a common Universal Time (00:36 UTC or 12:36 local time at the center of the image) and
encompasses local times from sunrise (west) to sunset (east) with all images rotated so that north is up.
In the LER scene, a large east-west belt of clouds are visible near the equator, as are cloud plumes
descending from the Arctic. The major cloud patterns change slowly, but show major seasonal changes.
Figure 5 shows six additional scenes from the same day, 17 April 2016, with large cloud features
associated with the Arctic region, an equatorial cloud band, and large cloud structures over the Antarctic
Ocean. Figure 6 shows reflectivity measurements for 23 November 2015 with cloud features common in
the Southern Hemisphere SH. The cloud band extending toward the Antarctic region from Argentina's
Salado River is an example of a persistent feature that appears frequently throughout the year.  In a
later section, the amounts of retrieved ozone and cloud reflectivity $0 < R_C < 1$ are used to estimate the
amount of UV radiation reaching the earth's surface over snow/ice free scenes.
The Arctic and Antarctic ice sheets are visible after their spring equinox times, and especially in
their respective late spring and summer images when the Earth's poles are tilted toward L-1 (Figs. 5 and
6). In the color and LER images, clouds over ice are not readily visible because of the very high ice
reflectivity providing little or no contrast with 388 nm cloud reflectivity.  It is possible to obtain
information about clouds over ice from the $O_2$ A-band channel at 764 nm (Fig. 7), which differentiates
between reflecting surfaces that are at different altitudes because of oxygen absorption in the
atmosphere. In this image, the bright white clouds (less atmospheric $O_2$ absorption) are at higher
altitudes than the grey clouds, which are all higher than the ice surfaces.  A quantitative analysis of cloud
height and cloud-caused reduction in solar irradiance reaching the ice surface will be the subject of a
future paper.

### 4    Validation of EPIC Ozone Retrieval


EPIC retrieved ozone can be validated by comparison with other ozone measuring satellite data
(e.g., OMI, and OMPS) and by comparison with well-calibrated ground-based instruments.
While EPIC observes from sunrise to sunset in every image, there are only 6 to 8 useful
coincidences per 24 hours with a specified ground site separated by either 68 minutes (NH summer) or
110 minutes (NH winter). Coincidences at high SZA > 75$^O$ are increasingly inaccurate for both satellite
and ground-based retrievals. This problem is compounded for EPIC, since high SZA also implies high SLA,
which increases the spherical geometry correction error. Ozone absorption and Rayleigh scattering at
high SZA also prevents 317.5 nm radiances from reaching into the lower troposphere and to the surface,
which is partially mitigated by having the retrieval algorithm automatically switch from 317.5 nm to 325
nm at high optical depths (usually high SZA).



A comparison of EPIC retrieved TCO with those determined by a Pandora spectrometer
instrument (#034) located at Boulder, Colorado is shown in Fig. 8. This Pandora was selected because it
has been extensively compared to a well calibrated Dobson spectroradiometer and to OMI and OMPS
ozone overpass data (Herman et al., 2015). The Pandora data are matched in location and time $t_O$ to the
EPIC UTC when Boulder, Colorado is in view (several times per 24 hours). Pandora ozone is averaged
over $t_O$ ±12 minutes. EPIC data are limited to distances within 50 km of Boulder, Colorado. Figure 8
shows that EPIC and Pandora ozone amounts track each other closely during 2015 and 2016. The 2015-
2016 average agreement is 2.7 ± 4.9 %. There is a period in the winter of 2016 where the Pandora data
quality was degraded by the presence of heavy cloud cover and in February by a mechanical problem
with the Pandora sun tracker.
The OMI and OMPS satellites are polar orbiting with an equator crossing time of about 13:30
hours local time measuring in a narrow strip on either side of the orbital track.  While it is possible to
compare EPIC ozone with low earth orbit satellite data, a more complete comparison can be made with
the assimilated ozone product from MERRA-2, the Modern-Era Retrospective Analysis for Research and
Applications, (Rotman et al., 2002) version 2  (MERRA-2, Molod et al., 2014). MERRA-2 ozone is based on
Microwave Limb Sounder (MLS) and total column ozone from the Ozone Monitoring Instrument OMI on
NASA's EOS *Aura* satellite. The advantage of using MERRA-2 is that the ozone field is synoptic and can be
directly compared with EPIC for the same UTC (Fig. 9) over the same sunlit globe as seen by EPIC. The ozone
structures seen by EPIC are all present in the MERRA-2 independent assimilation, even though there is an
average offset of about 10 DU (3 %). The disagreement with EPIC is similar to the offset of MERRA-2 with
other satellite data (Wargan et al., 2017). A close look at the ozone maps in Fig. 9 shows overall agreement
with most features including the small region of elevated $O_3$ over the central US. There are differences, such
as the higher amount of $O_3$ measured by EPIC over Brazil on 23 November and the structure at $15^O$N in the
transition from equatorial $O_3$ values to mid-latitude values (dark blue to light blue).

**5.0 Synoptic Variation of Ozone (SVO) from Sunrise to Sunset**

Most LEO satellite views of ozone are at almost fixed local time based on the equator crossing
local solar time (13.5 ± 0.8 hours side scanning) with approximately 20 minutes local time variation from
the equator to the pole.  Longitudinal coverage is obtained by piecing together North-South strips
obtained about 90 minutes apart.  Variation that occurs on a scale less than 90 minutes cannot be seen
from a polar orbiting LEO satellite, nor can variation from different local times of the day. EPIC observes
from close to sunrise and sunset with local solar noon near the center of the data set as shown in Fig.
10. The exact position of noon in the EPIC images depends on the location of EPIC in its orbit relative to
the Earth-Sun line. The longitude resolution is approximately $0.25^O$ at the center of the FOV, which
corresponds to a time resolution of about 1 minute. The resolution decreases as the secant of the angle
from the center (e.g., 2 minutes or $0.5^O$ at $60^O$ from the center).  A limitation in the EPIC observations
occurs at high SZA and high SLA. As can be seen in Fig. 10, ozone values near the morning terminator are
probably too low compared to the middle longitude values. These retrieval errors are partly caused by
the effects of spherical geometry that are not properly represented in the TOMRAD radiative transfer
calculations.





The view of the EPIC instrument from sunrise to sunset at fixed UTC is not the diurnal variation
that an instrument on the ground would see from sunrise to sunset.  For the ground-based Pandora
instrument, the observed changes throughout the day from sunrise to sunset are at varying UTC every
80 seconds.  Compared to the ground-based viewpoint, EPIC obtains data for a fixed geographic location
every 68 minutes UTC in NH daytime summer and every 110 minutes in NH daytime winter.
**5.1 Southern Hemisphere SH Late Spring 23 November 2015** :
To illustrate the SH synoptic change in ozone, Figs. 10 and 11 show the diurnal (longitudinal)
variation of ozone centered on the South American continent on 23 November 2015 at 16:20 UTC. The
local time varies from early morning (06:20, -150$^O$ longitude) to late-afternoon (16:20, 0$^O$ longitude).  At
high southern latitudes, 60$^O$S and 70$^O$S, the late spring (23 November) residue of 2015 Antarctic ozone
hole is clearly visible in the ozone map image (Fig. 10). Figure 11 shows details of the ozone amounts in
specified latitude bands (±0.125$^O$ wide) in the Southern Hemisphere sampled every 5$^O$ degrees from 0$^O$
to 70$^O$S.  Solar zenith angles are limited to the range ±70$^O$ to avoid high latitudes and longitudes near
sunrise or sunset where spherical geometry effects become important. This particular example (Fig. 11)
is from one image centered over South America (Fig. 10). For 23 November there are 15 more
overlapping images covering the entire 360$^O$ of longitude that could be combined to produce a complete
composite global map of ozone at 15 different UTCs. In the NH summer there would be 22 images per
day.  A composite ozone map of this kind would no longer be synoptic, since overlapping data are
averaged, but would now be similar to the joined data strips from OMI or OMPS.
Figure 11 contains the data points from a 0.25$^O$ x 0.25$^O$ average within each 5$^O$ latitude band L
shown as light grey dots. The dark lines are a Lowess(0.05) fit (locally weighted least squares fit to 5 % of
the data, (Cleveland, 1981)), which corresponds to approximately a 30 minute time average (7.5$^O$
Longitude). The largest apparent scatter from the Lowess fit occurs at L = 0.125$^O$S, which amounts to a
longitudinal standard deviation from the mean of ±4 DU or ±1.5 %.The equatorial bands (0$^O$S to 20$^O$S)
shows considerable longitudinal change (10–20 % from L = 0–40$^O$S rising to 75 % at L = 70$^O$S,
approximately as TCO = 16.063 + 0.56L + 0.02L$^2$). Most of the observed changes are dynamically driven,
since the photochemistry involved in the stratosphere (20 - 25 km altitude) is too slow to produce such
large changes with changing SZA. Southward of 45$^O$S, the effects of the remaining ozone hole depletion
(dark blue in Fig. 10), which is still present in November, appear at -50$^O$ longitude as indicated in Fig. 11.
**5.2 Northern Hemisphere NH Summer Solstice 21 June 2016:**
An example is provided for the ozone retrievals obtained on 21 June 2016 at 18:41 UTC that is
approximately centered over North America (Fig. 12). Since this is Northern Hemisphere summer
solstice, corresponding to the sun being nearly overhead at 23$^O$N, the latitude range available for
retrieving ozone extends over the North Pole. Figure 13 contains ozone retrievals in 0.25$^O$ wide latitude
bands similar to Fig. 11.  Unlike the SH 23 November 2015 example, there is only moderate longitudinal
(diurnal) variability in ozone amount for latitudes between 0$^O$ and 15$^O$N. However, there is a clear wave
structure in the 20$^O$N to 25$^O$N bands with a periodicity of approximately 35$^O$ longitude (2.3 hours) and
again in the 40$^O$N to 60$^O$N bands that are not obvious in the global map (Fig. 12).



The dynamical effects on ozone in the NH mid-latitudes are quite different than their
counterparts in the SH, where the NH mid-latitude behavior (30°N–35°N) is clearly separated from
equatorial and high latitude bands with an increase in ozone amount from about 280 DU to about 350
DU, which is larger than a similar increase in the SH. There is an ozone periodicity of approximately 38°
longitude (2.5 hours) at 30°N–35°N midday and a longer longitudinal period 73° (4.9 hours) in the
morning. At higher latitudes, 35°N–55°N, the variability is more pronounced with an approximate
period of 55° (3.6 hours). In the bands from 55°N–70°N the variability is reduced and the ozone amount
falls from mid-latitude values of about 350 DU to below 300 DU. The wave structure varies throughout
the year in both hemispheres.
**5.3 Northern and Southern Hemisphere 17 April 2016 18:35 UTC**
Figure 5-5 shows the ozone retrieval for the sunlit globe on 17 April 2016 at 18:36 UTC about 1
month from the March equinox including large plumes of elevated ozone amounts (450 DU) extending
from high latitudes into mid-latitudes where the usual ozone amount is about 350 DU. For the SH (Fig. 5-
5), polar ozone variability (280-320 DU) is relatively small compared to November 23 (Fig. 10).  There is
wave structure (Fig. 15) between 30°S and 40°S with a periodicity of about 4 hours (60° longitude) (see
also Schoeberl and Kreuger, 1983). The dip in $O_3$ amount at 77°W to 67°W and 10°S to 25°S
corresponds to the Andes Mountains in Peru, Bolivia, and Chile. While the SZA range is limited to ±70°,
the SLA reaches more than 80° at low latitudes for longitudes between 40°S and 20°S introducing
spherical geometry correction errors that increase towards sunset near 20°W. The errors appear as
apparent increases in $O_3$ amount. At higher latitudes, the SLA is in the middle 70°s when the SZA is 70°.
The high SLA error is present in both hemispheres for observations near equinox.
The NH shows little variability in the equatorial region (0–25°N) with a mean value of about 260
DU (Fig. 16). The SLA error is present for latitudes between 0 and 15°N and 0 and 15°S that appears as
an elevated ozone amount at longitudes east of 50°W. Mid-latitudes (30°N to 40°N) show a wave
structure that is approximately 37° apart (2.5 hours) at 35°N. A similar structure occurs in the SH with a
period of about 4.5 hours. There is an ozone maximum (red area in Fig. 14 about 450 DU) near 140°W
extending from 60°N to 35°N, very high ozone amounts in the Arctic region, and a high ozone patch
over the central US (35°N to 45°N and 104°W) peaking at 420 DU (40°N and 104°W), which probably
corresponds to a region of high atmospheric pressure.
**6.0  Estimating Erythemal Irradiance at the Earth's Surface**
The unique observing geometry of DSCOVR/EPIC permit the use of  synoptic ozone and cloud
reflectivity data to be used to compute the diurnal variation of UV irradiance from sunrise to sunset for
any point on the illuminated earth observed by EPIC.  Previous calculations from satellite data used
cloud cover and ozone from 13:30 and assumed it applied to local noon. The assumption is usually
adequate for slowly varying ozone, but not for estimating the effects of more rapidly varying cloud
cover. The following paragraphs discuss the calculation of erythemal irradiance, a spectrally weighted
mixture of UV wavelengths used as a measure of skin reddening and potential sunburn from exposure to
sunlight.



Erythemal irradiance $E_0$(SZA $\theta$, altitude Z) at the earth surface (watts/m$^2$) is defined in terms of
a wavelength dependent weighted integral over a specified weighting function A($\lambda$) times the incident
solar irradiance I($\lambda,\theta,\Omega,C_T$) (Watts/m$^2$) (Eq. 11) at the Earth's surface. The erythemal weighting function
Log$_{10}$(A$_{ERY}$($\lambda$)) is given by the standard Erythemal fitting function shown in Eq. 12 (McKinley and Diffey,
1987). Tables of radiative transfer solutions for D$_E$ = 1 AU are generated for a range of sza (0 < $\theta$ < 90$^0$),
for ozone amounts 100 < $\Omega$ < 600 DU, and terrain heights 0 < Z < 5 km using the TUV DISORT radiative
transfer model as described in Herman (2010) for erythemal and other action spectra (e.g., plant
growth, vitamin D production, cataracts, etc.).

$$E_0(\theta,\Omega,C_T) = \int_{250}^{400} I(\lambda,\theta,\Omega,C_T)A(\lambda)d\lambda \qquad (11)$$

| | | |
|---|---|---|
| 250 < $\lambda$ < 298 nm | Log$_{10}$(A$_{ERY}$) = 0 | (12) |
| 298 < $\lambda$ < 328 nm | Log$_{10}$(A$_{ERY}$) = 0.094 (298 - $\lambda$) | |
| 328 < $\lambda$ < 400 nm | Log$_{10}$(A$_{ERY}$) = 0.015 (139 - $\lambda$) | |

Equation 11 can be accurately approximated by the power law form (Eq. 13), where U($\theta$) and R($\theta$)
are fitting coefficients to the radiative transfer solutions in the form of rational fractions. Rational
fractions were chosen because they tend to behave better at the ends of the fitting range than
comparable fitting accuracy polynomials.

$E_0(\theta,\Omega,C_T)$ = U($\theta$) ($\Omega$/200)$^{-R(\theta)}C_T$      (13)

U($\theta$) or R($\theta$) = (a+c$\theta^2$+e$\theta^4$)/(1+b$\theta^2$+d$\theta^4$+f$\theta^6$)   $r^2$ > 0.9999      (14)

$C_T$ = (1-LER)/(1-R$_G$) where R$_G$ is the reflectivity of the surface      (15)

E($\theta,\Omega$,Z) = $E_0(\theta,\Omega)$ H($\theta,\Omega$,z)      (16)

H($\theta,\Omega$,Z) = 1+(0.04652 Z$_{km}$ +0.00496) (-0.07033 ($\Omega$/200) + 1.12303)G($\theta$ )      (17)

G($\theta$) = g+h$\theta$+i$\theta^2$+j$\theta^3$+k$\theta^4$      (18)

The coefficients  a, b, c, d, e, f, g, h, j, and k are in Tables A-1 and A-2 in the appendix

The E$_0$ solutions to the radiative transfer calculations can be accurately reproduced by a relatively
simple functional form (Eqs. 13 to 15) with the coefficients given in Table A-1. These are the same
coefficients given in Herman (2010) along with other biological action spectra weighting functions, H(z,$\theta$)
is a function representing the increase in E($\theta,\Omega$,Z) with altitude per km, and C$_T$ is the cloud transmission
function (Eq. 15) estimated from the retrieved LER derived by assuming that the cloud-ground system
can be approximated by a two-layer Stokes problem (elevated cloud and surface) with atmospheric
effects between the cloud bottom and the surface neglected (Herman et al.,2009).  $r^2$ is a measure of
the correlation of the E$_0$ data points with the fitting function. Eqs. 13 to 18 are for an Earth-Sun distance
of 1 AU.



451  For $E_0$ The fitting residual is less than ± 0.001 W/m$^2$ compared to the worst case when $E_0$(50$^o$,

452 200) = 0.15 W/m$^2$ (Herman, 2010). When height effects are included $E(\theta,\Omega,Z)$ = $E_0(\theta,\Omega)$ $H(\theta,\Omega,Z)$,

453 where $H(\theta,\Omega,Z)$ is a fitting polynomial (Eq. 17) to the downward irradiance at 0, 1, 2, 3, 4, and 5 km

454 based on results from the radiative transfer calculation. The increase of erythemal irradiance with

455 altitude has an SZA dependence given by $G(\theta)$, which increases with $\theta$ until $\theta$ is approximately 60$^o$, and

456 then $G(\theta)$ decreases.

457  The height dependence of $E(\theta,\Omega,Z)$ is similar to that derived by Chubarova et al. (2016) for low

458 aerosol amounts. When absorbing aerosols have a significant optical depth, Chubarova et al. (2016)

459 derived a multiplicative correction term to $E(\theta,\Omega,Z)$ for a wide variety of conditions.

460

461  When Eq. 13 is applied to the ozone and LER data described in previous sections, the global

462 erythemal irradiance at the ground can be obtained after correction for the Earth-Sun distance $D_E$ in a

463 manner similar to Eq. 1, where $D_E$ in AU can be approximated by (Eq. 19),

464

   $D_E$ = 1 – 0.01672 cos(360 (day_of_year – 4)/365.25)     (19)

465

466  An example of $E(\theta,\Omega,Z)$ is shown in Fig. 17 for 17 April 17 2016 at 18:35 UTC. Local noon is near

467 the center of the image with sunrise to the left (west) and sunset to the right (east). For this date, the

468 sun is overhead just north of the equator producing very high values of Erythemal irradiance $E(\theta,\Omega,Z)$

469 corresponding to a UV index, UVI, of 13 at sea level in the Pacific Ocean (UVI = 40 $E(\theta,\Omega,Z)$). The UVI

470 scale was designed for sea level mid latitudes ranging from 0 to 10 to provide public health warnings

471 (e.g. for UVI = 8). Somewhat higher values are seen in the Sierra Nevada Mountains in Mexico near

472 20$^O$N. This particular day is relatively cloud free over most of South America except for clouds over

473 southern Brazil extending into Paraguay and other small patches of clouds. For the erythemal irradiance,

474 the presence of clouds reduces the amount of UV reaching the ground (blue color with a UV index of

475 less than 4).

476

477  The increase with altitude is much more pronounced during the summer months over the Andes

478 Mountains reaching above 4 km (over 13,000 feet). Figures 18 and 19 show the large increases with

479 altitude over the Andes Mountains for 23 November 2015, with the sun nearly overhead at 20$^O$S

480 latitude. Here the UV index ranges from 16 to 18, which agrees with previous ground-based

481 measurements in this region (Cede et al., 2002). Any significant unprotected exposure to these levels of

482 UV would lead to severe sunburn and eye damage. On a completely clear day the UV index would be

483 even higher than 18. Figure 19 is a longitudinal slice through the UV data in Fig. 18 at 20$^O$S. The figure

484 shows the longitudinal variation $E(\theta,\Omega,Z)$ as a function of local time, the effect of light clouds on the

485 eastern side of the Andes Mountains, and the sharp reduction at 50$^O$W.

486

487  Figure 20 shows the erythemal irradiance computed for 21 June 2016 centered over the US and

488 Central America. The sun is overhead at 23.3$^O$N latitude. In the clear regions not covered with light

489 clouds, the UV index reaches about 12 extending from an area in the Pacific Ocean at 15$^O$N up into the

490 US mid-west, Rocky Mountains, Utah and New Mexico. The eastern US has a lower UV index of about 8.



The extended scale of this map (UVI = 0 to 20) is too coarse to see the variation with latitude on the east
coast.

Similarly, Fig. 21 shows high values of Erythemal irradiance in the Himalayan Mountains on June
21, 2016 with peak UV index of about 15 even in the presence of partial cloud cover that reflects a
portion of the incident solar flux back to space. The effect of cloud cover can be seen in Fig. 22, which is
a longitudinal slice through the irradiance values associated with the latitude at $32^O$N. In the absence of
clouds, the peak value of the UV index would be close to 20. Even with cloud cover, the UV index
reached 15, which is twice the value of a typical cloudless summer case in the US at comparable
latitude.
**7.0 Summary**
The DSCOVR/EPIC 10-filter Spectroradiometer (317.5 to 780 nm) makes measurements of the
the rotating sunlit face of the earth from the Lagrange-1 point located $1.5 \times 10^6$ km from the earth with a
maximum resolution of 10 x 10 $km^2$ for 443 nm at the sub-satellite point. The other 9 channels have 18 x
18 $km^2$ resolution. The key difference between EPIC and LEO satellites is EPIC's ability to measure the
whole sunlit earth (sunrise to sunset) at the same UTC (synoptic measurements) every 68 or 110
minutes depending on the season at the Wallops Island, Virginia data receiving station. EPIC ozone
retrievals have been compared successfully to both ground-based Pandora spectrometer instruments
and to the MERRA-2 satellite data assimilation model for the same UTC observed by EPIC.   EPIC's
synoptic measurements insure that the ozone amounts, cloud reflectivity, and aerosol amounts that are
used to estimate UV irradiance are the proper values for each time of the day. EPIC has been making
measurements since June 15, 2015 with no evidence of significant degradation relative to LEO satellites
observing the same scene at the same angles. EPIC has obtained ozone and reflectivity data multiple
times per 24 hours for over two years that can be used to more accurately estimate the health effects
from continuous or periodic exposure during any day to UV radiation reaching the ground including the
effects of cloud cover and altitude.












**Appendix**
Figure A1 illustrates the orbit of the DSCOVR spacecraft following the earth in its orbit about the
sun.

Table A-1   Coefficients R($\theta$) and scaling coefficient U($\theta$) for  $0 < \theta < 80^O$
and  $100 < \Omega < 600$ DU   for $E(\Omega,\theta) = U(\theta)\,(\Omega/200)^{-R(\theta)}$   ( 1.0E10 = $1.0 \times 10^{10}$)

U($\theta$) or R($\theta$) = $(a+c\theta^2+e\theta^4)/(1+b\theta^2+d\theta^4+f\theta^6)$   $r^2 > 0.9999$

Action Spectra     U($\theta$) (watts/m$^2$)                                  R($\theta$)

CIE Erythemal      a= 0.4703918683355716              a= 1.203020609002682
$U_{ERY}$ & $R_{ERY}$   b= 0.0001485533527344676          b= -0.0001035585455444773
                   c= -0.0001189976502179551          c= -0.00013250509260352
                   d= 1.915618238117361E-08            d= 4.953161533805639E-09
                   e= 7.693069873238405E-09            e= 1.897253186594168E-09
                   f= 1.633190561844982E-12            f=  0.0

Table A-2 Solar Zenith angle function G($\theta$) used in Eq. 18
          G($\theta$)   = $g+h\theta+i\theta^2+j\theta^3+k\theta^4$

          g= 0.9996074048174048          j= 1.412462444962443E-06
          h= 0.0001453776871276851        k= -2.037907925407924E-08
          i= 2.806514180264192E-05















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



**Tables**

Table 1 Exposure Times for viewing the Earth and Full Moon (Earth side view)

| Wavelength | Earth Exposure (ms) | Full Moon Exposure(ms) | Filter Width (nm FWHM) |
|---|---|---|---|
| 317.5 | 654 | 2500 | 1 |
| 325 | 442 | 500 | 1 |
| 340 | 67 | 92 | 3 |
| 388 | 87 | 95 | 3 |
| 443 | 28 | 100 | 3 |
| 551 | 22 | 70 | 3 |
| 680 | 33 | 105 | 1.7 |
| 688 | 75 | 224 | 0.6 |
| 764 | 101 | 250 | 1.7 |
| 779.5 | 49 | 180 | 2 |








Table 2  $\pi K_M$ on 1 January 2016                 Irradiance at 1 AU

| M | $\lambda$ (nm) | $\pi K_{MO}$ | $S_M(mW/m^2/nm)$ |
|---|---|---|---|
| 1 | 317.478 | 1.216E-04 | 819.0 |
| 2 | 325.035 | 1.111E-04 | 807.7 |
| 3 | 339.858 | 1.975E-05 | 995.8 |
| 4 | 387.923 | 2.685E-05 | 1003. |





Table A1   Coefficients R(θ) and scaling coefficient U(θ) for $0 < \theta < 80^O$
and $100 < \Omega < 600$ DU   for $E(\Omega,\theta) = U(\theta) \, (\Omega/200)^{-R(\theta)}$   ( 1.0E10 = 1.0x10$^{10}$)

$$U(\theta) \text{ or } R(\theta) = (a+c\theta^2+e\theta^4)/(1+b\theta^2+d\theta^4+f\theta^6) \quad r^2 > 0.9999$$

| Action Spectra | $U(\theta)$ (watts/m$^2$) | $R(\theta)$ |
|---|---|---|
| CIE Erythemal $U_{ERY}$ & $R_{ERY}$ | a= 0.4703918683355716<br>b= 0.0001485533527344676<br>c= -0.0001188976502179551<br>d= 1.915618238117361E-08<br>e= 7.693069873238405E-09<br>f= 1.633190561844982E-12 | a= 1.203020609002682<br>b= -0.0001035585455444773<br>c= -0.00013250509260352<br>d= 4.953161533805639E-09<br>e= 1.897253186594168E-09<br>f= 0.0 |





Table A2 Solar Zenith angle function G(θ) used in Eq. 18

$$G(\theta) = g + h\theta + i\theta^2 + j\theta^3 + k\theta^4$$

g= 0.9996074048174048           j= 1.412462444962443E-06

h= 0.0001453776871276851     k= -2.037907925407924E-08

i= 2.806514180264192E-05





**Figure Captions**

f01 Filter transmission functions (percent) for the 10 EPIC wavelengths

f02 Normalized calibration functions referenced to its value at 4 Jan 2016 when $D_E$ = 1 au. Average rate of increase is 0.016 per year.

f03 Natural Color EPIC Earth images from June 6 and December 6, 2016 showing the field of view during the respective hemispheric summers.  In both of these images, 6-months apart, the EPIC orbit is to the west of the Earth-Sun line causing the west side of the globe (sunrise) to appear brighter than the east side (sunset). Notice the bright specular reflection over Argentina, South America embedded within a cloud feature. This is thought to be from ice crystals in high clouds (Marshak et al., 2017).

f04 EPIC retrieved ozone and LER values for April 17, 2016 at 00:36 UTC.  The ozone scale is from 100 to 500 DU, and the LER scale is from 0 to 100 percent.

f05 LER at six sequential UTC 0:36, 2:24, 4:12, 6:00, 7:48, and 9:36 from 17 April 2017 showing clouds in the arctic region as the earth rotates in EPIC's field of view.

f06 Cloud formations from 23 Nov 2015 showing cloud cover in the Southern Hemisphere and near Antarctica at 6 different UTC's, 10:56, 12:44 14:32, and 16:20, 14:32, 18:09, and 19:57.

f07 $O_2$ A-band View of Antarctica on December 6, 2015 showing clouds over ice. The white bright clouds are at higher altitudes than the dull grey clouds because of a combination of less oxygen absorption and higher optical depth.

f08 Daily $O_3$ data for EPIC (red) and Pandora (Grey) 2015 - 2016.  Left: EPIC ozone data compared to Pandora retrievals at Boulder Colorado. Right: Percent difference between EPIC and Pandora.

f09 Comparison of EPIC total column ozone with the MERRA-2 assimilation model ozone.

f10 Global image of ozone field for Fig. 11 for 23 Nov 2015 at 16:20 UTC

f11 Longitudinal or diurnal variation of ozone for the Southern Hemisphere every $5^O$ degrees from $0^O$ to $70S^O$ for 23 Nov 2015 at 16:20 UTC. The grey points are the individual data points in the band.  The solid lines are a Lowess(0.05) fit to the data points representing a solar time average from 0.6 to 0.7 hours depending on latitude. The SZA is limited to $\pm70^O$. Longitude = 0 Corresponds to 16:20 local time and longitude = -150 corresponds to 06:20 local time.

f12 Global image of ozone field for Fig. 13 for 21 June 2016 at 18:41 UTC

f13 Longitudinal or diurnal variation of ozone for the Northern Hemisphere every $5^O$ from $0^O$ to $70^O$ for 21 June 2016 at 18:41 UTC. The grey bands are the individual data points in the band.  The solid lines are a Lowess(0.05) fit to the data points representing a solar time average from 0.6 to 0.7 hours depending on latitude.  The SZA is limited to $\pm70^O$. Longitude = 0 Corresponds to 18:41 local time and longitude = -180 corresponds to 06:41 local time.





f14 Global image of ozone field for Figs. 15 and 16 for 17 April 2016 at 18:36 UTC.
f15 Southern Hemisphere: Solid lines are approximately 30 minute averages in solar time at 18:38 UTC
on 17 April 2016 for ozone variation between $0^O$ and $55^O$S latitude in $0.25^O$ latitude bands for 17 April
2016 at 17:36 UTC.
f16 Northern Hemisphere: Solid lines are approximately 30 minute averages in solar time at 18:38 UTC
on 17 April 2016 for ozone variation between $0^O$ and $75^O$N latitude in $0.25^O$ latitude bands for 17 April
2016 at 17:36 UTC.
f17 Erythemal irradiances calculated from Eq. 13 and from the EPIC ozone and LER data obtained on
April 17, 2016 at 18:35 UTC.  The scale shows both the irradiance values in W/m2 and the UV index
ranging from 0 to 20. This scene is centered over the Pacific Ocean and shows a peak UV index of about
15. Since this period is close to equinox, the sun is nearly overhead just north of the equator with solar
noon at $98.75^O$W longitude and overhead near $10^O$N.
f18 Erythemal irradiances centered over South America on November 23, 2015 at 16:19 UTC showing
extremely high values in the Andes Mountains in Peru, Bolivia, and Chile corresponding to a UV index
greater than 20. Local solar noon is at $64.75^O$W and overhead near $20^O$S.
f19 Erythemal Irradiances in a longitudinal slice at $20^O$S through a peak occurring in the Andes
mountains. Local noon is at $64.75^O$W.
f20 Erythemal irradiances centered over the United States on June 21, 2016 showing high values over
the Rocky Mountains and a portions of the Sierra Nevada Mountains.  The UV index reaches about 15.
Local solar noon is at $99.75^O$W and overhead near $23.3^O$N.
f21 Erythemal UV irradiances centered over the Indian Ocean on June 21, 2016 showing high values over
the Himalayan Mountains with the UV index exceeding 14.  UV levels are moderated by partial cloud
cover reflection of radiation back to space.  Solar noon is at $80.25^O$E.
f22 Erythemal Irradiances in a longitudinal slice at $32^O$N through a portion of the Himalayan mountains.
Local solar noon is at $80.25^O$E.
fA1 An illustration of DSCOVR's Lagrange-1 orbit





**Figures**

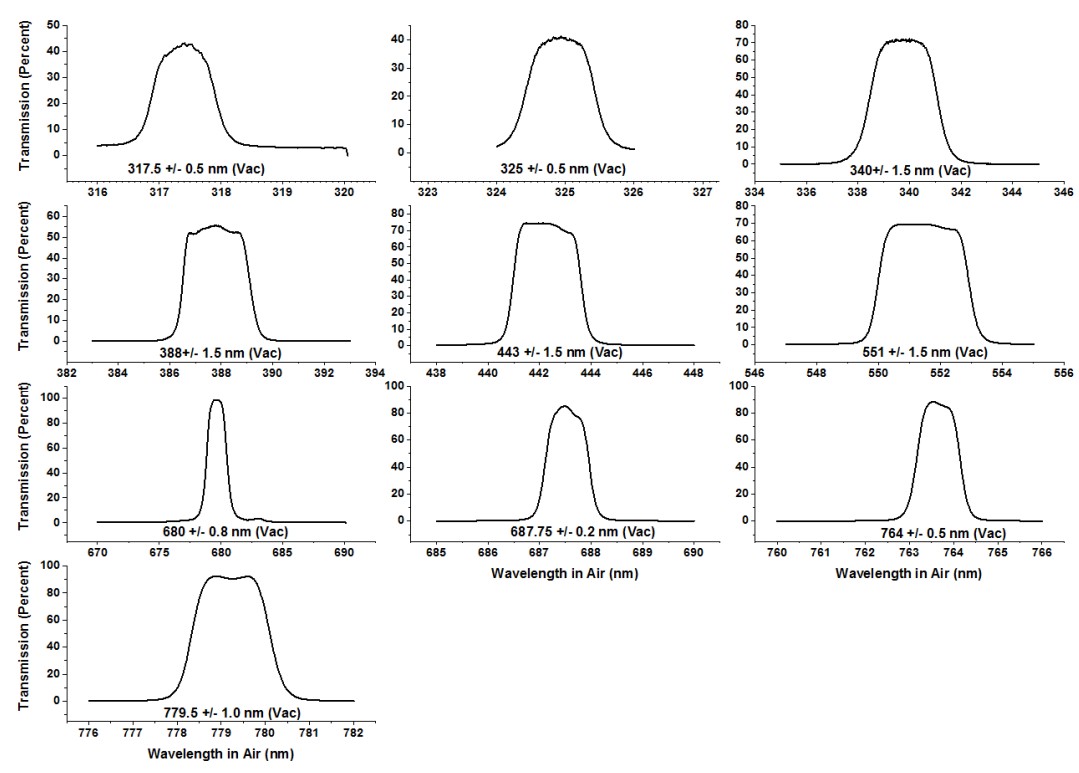

**f01**




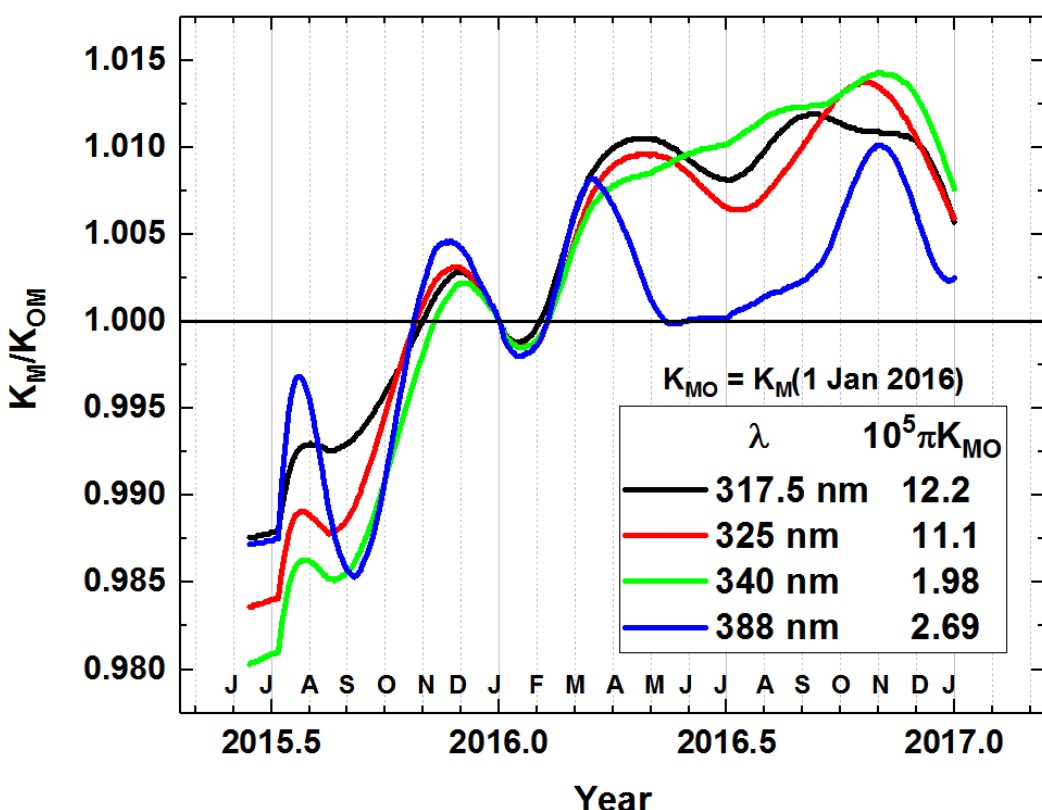

**f02**





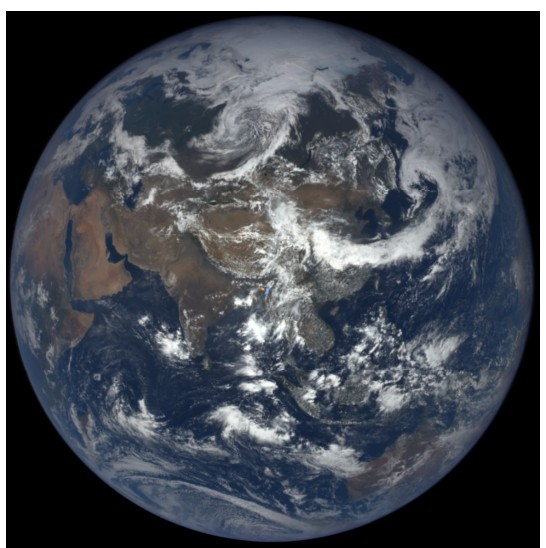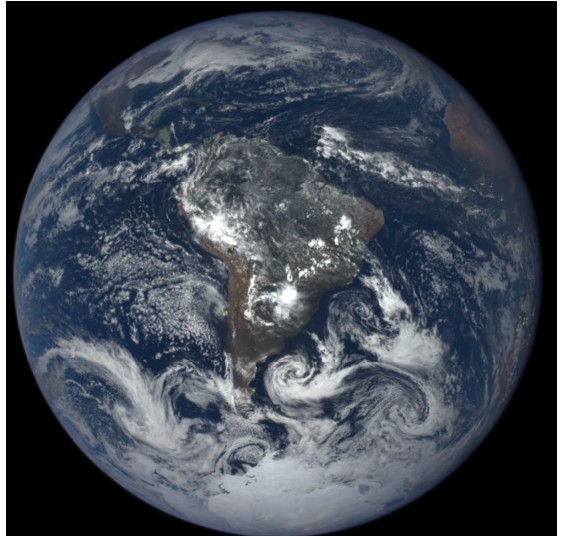

**f03**






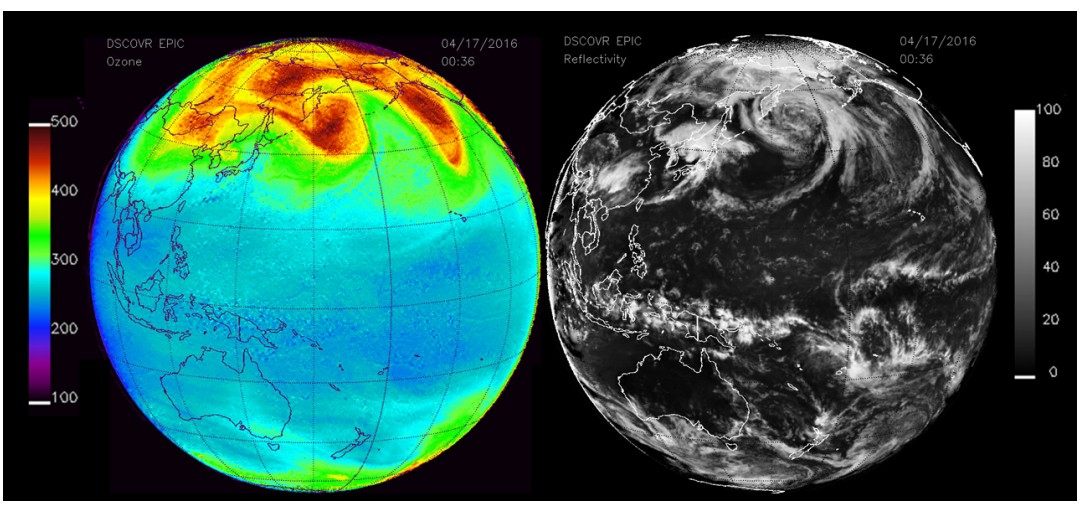

**f04**







**f05**



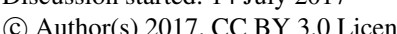

f06





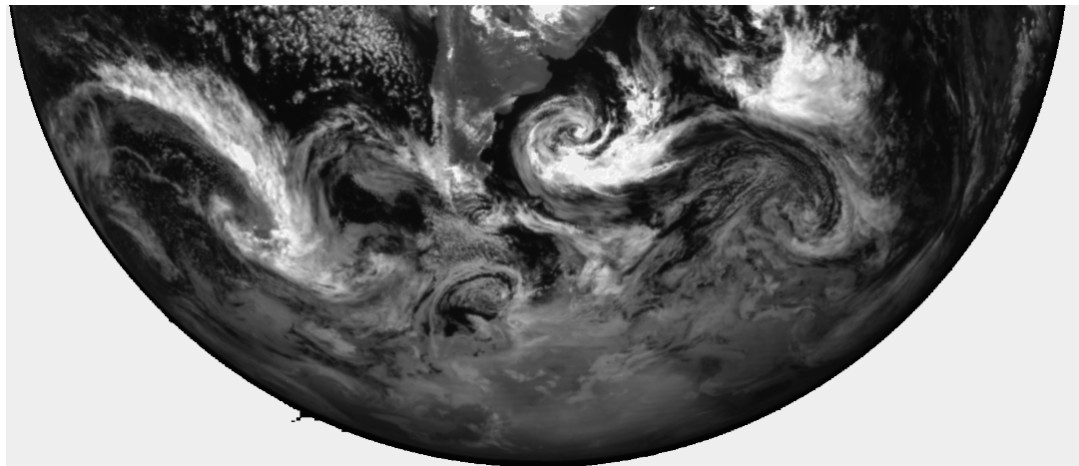

**f07**










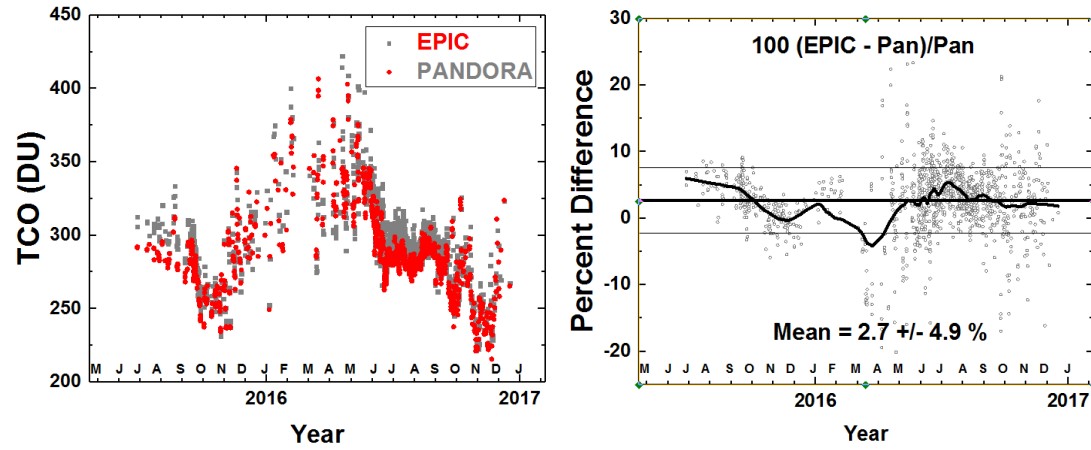

**f08**





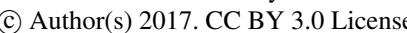


**f09**




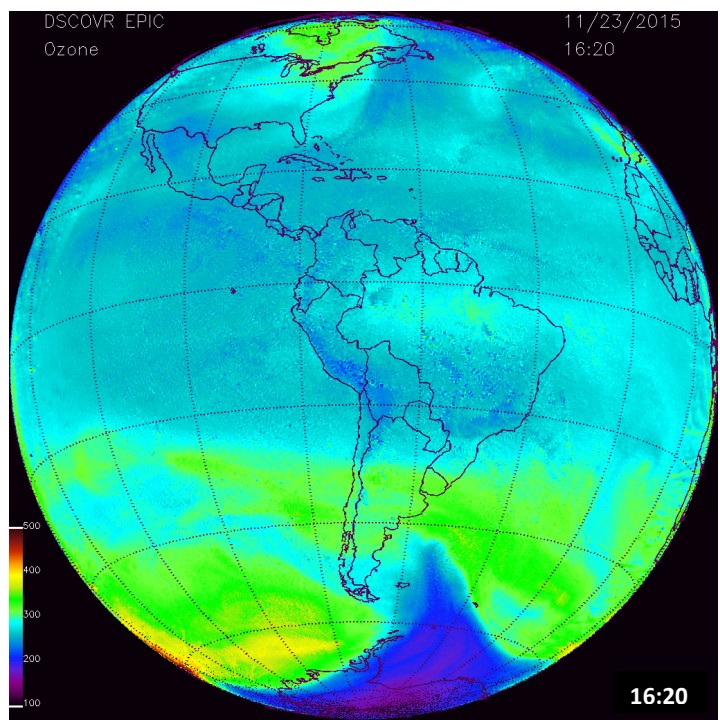

**f10**





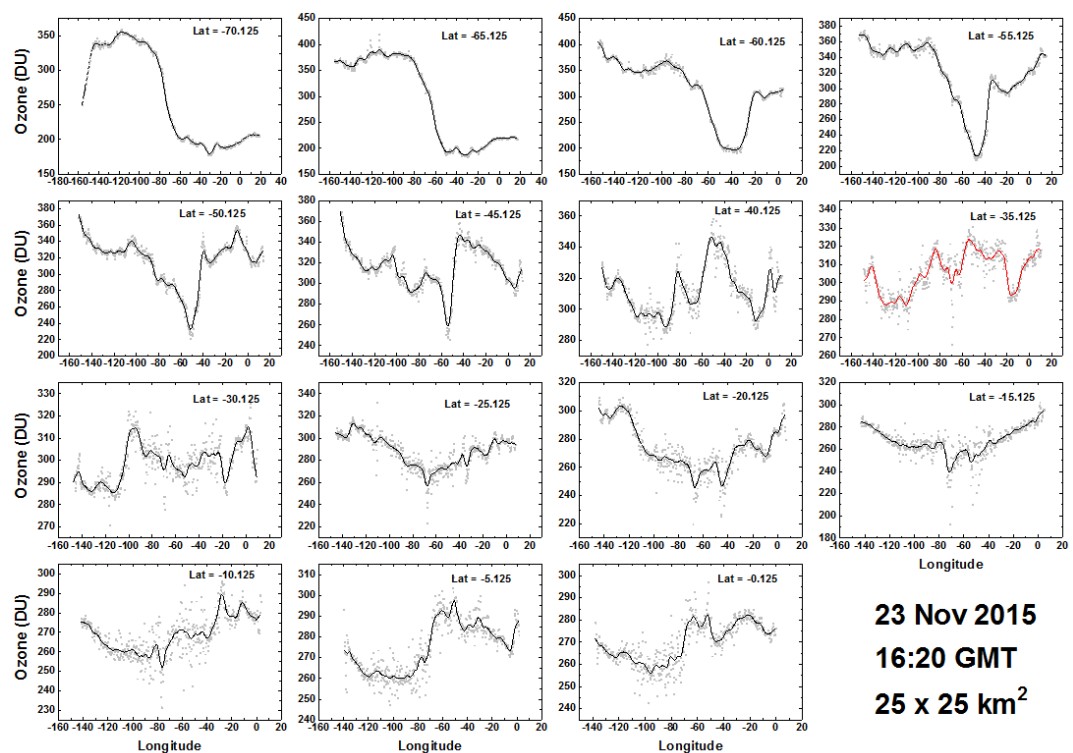

**f11**






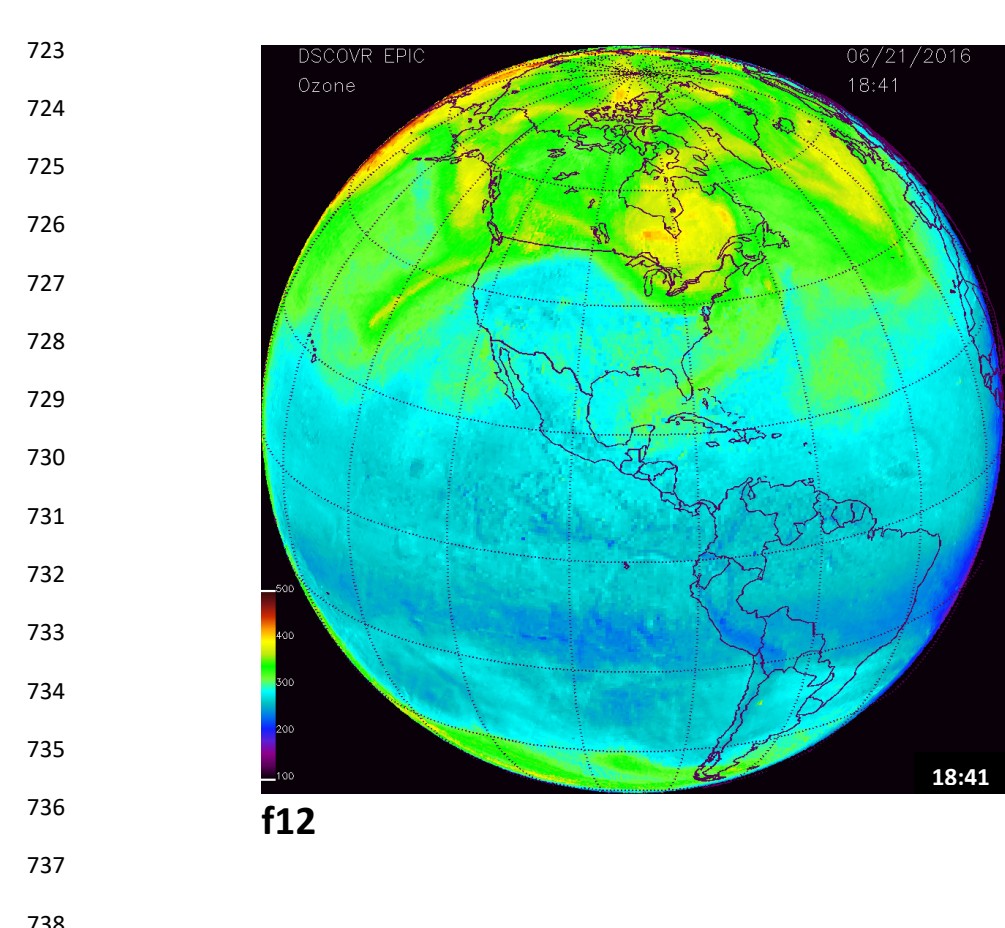

**f12**




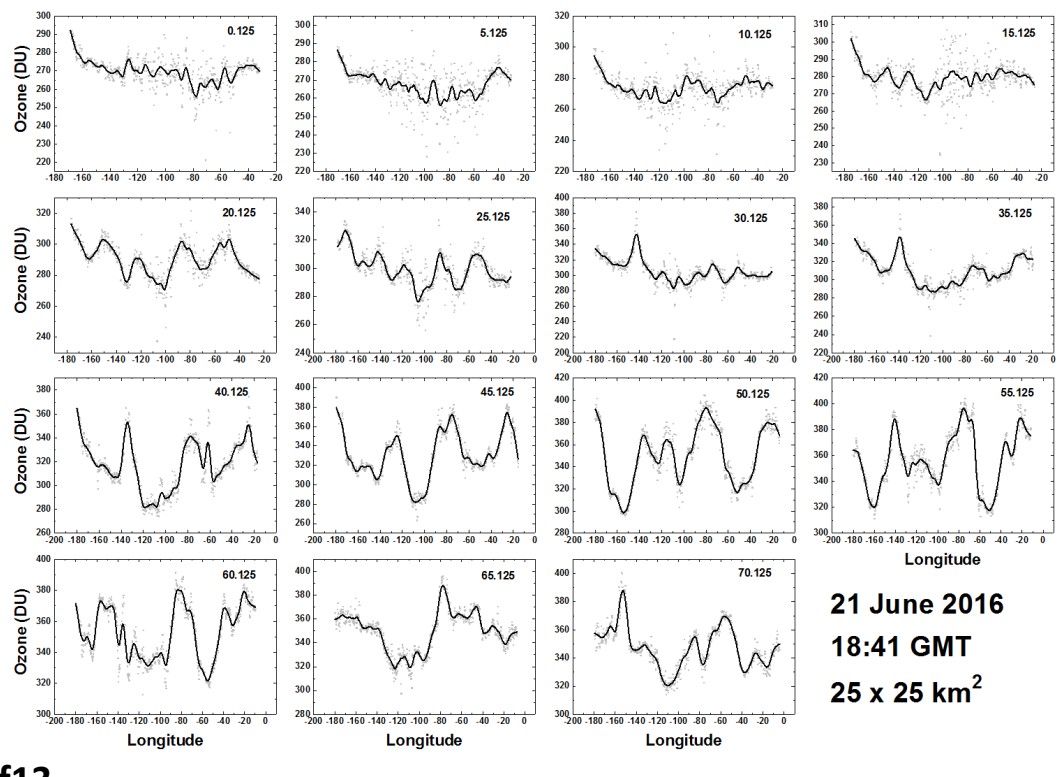

**f13**















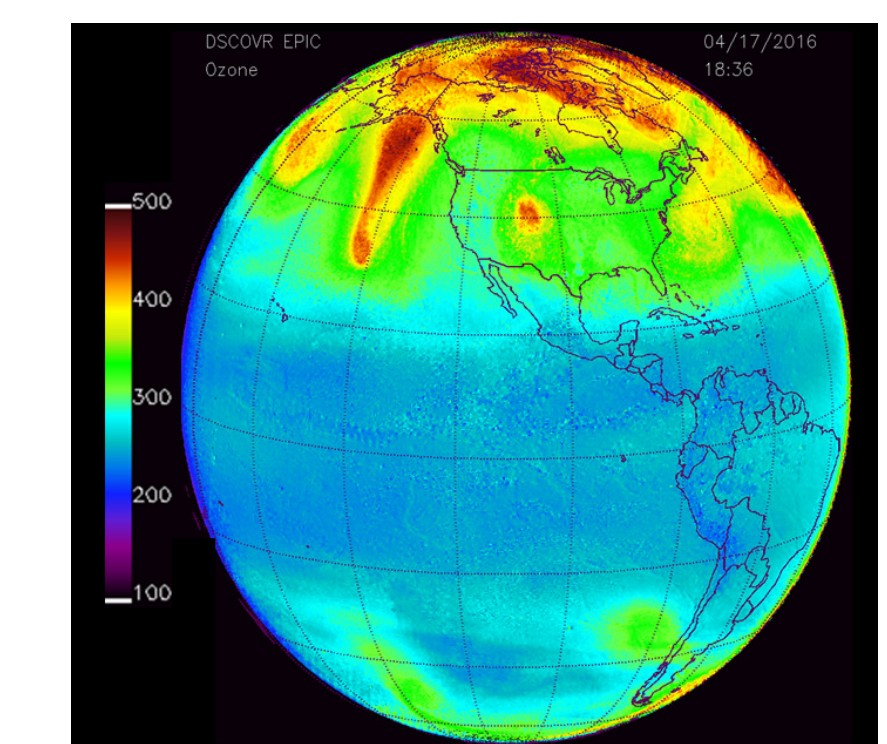

**f14**





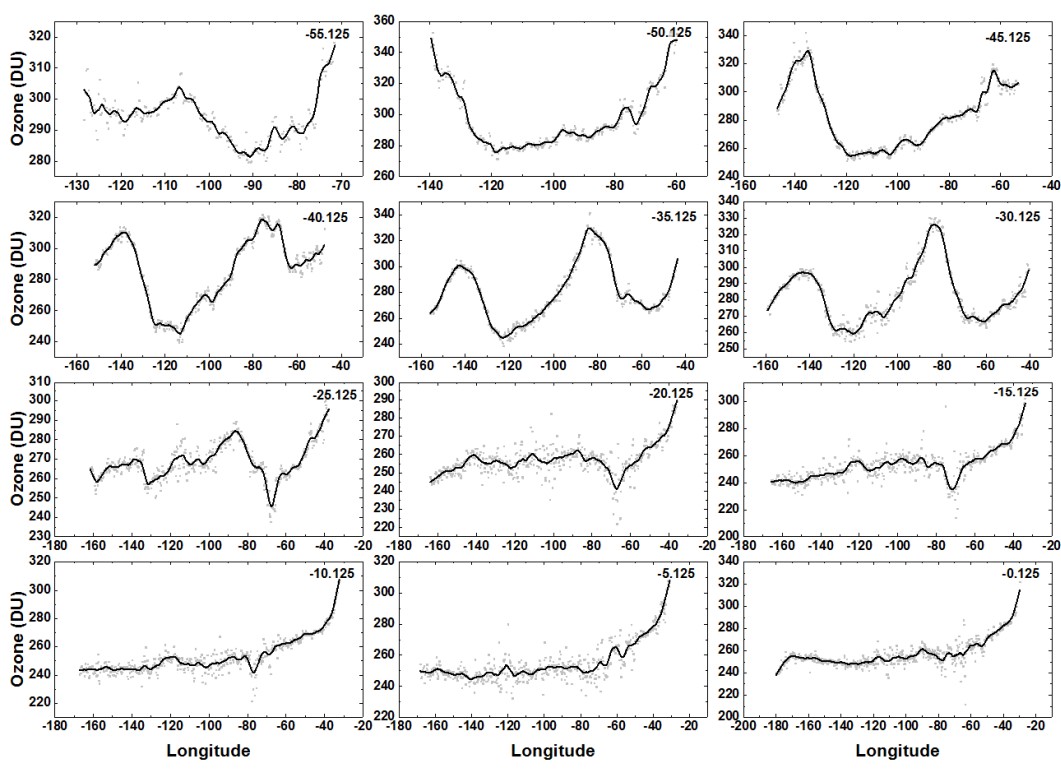

**f15**




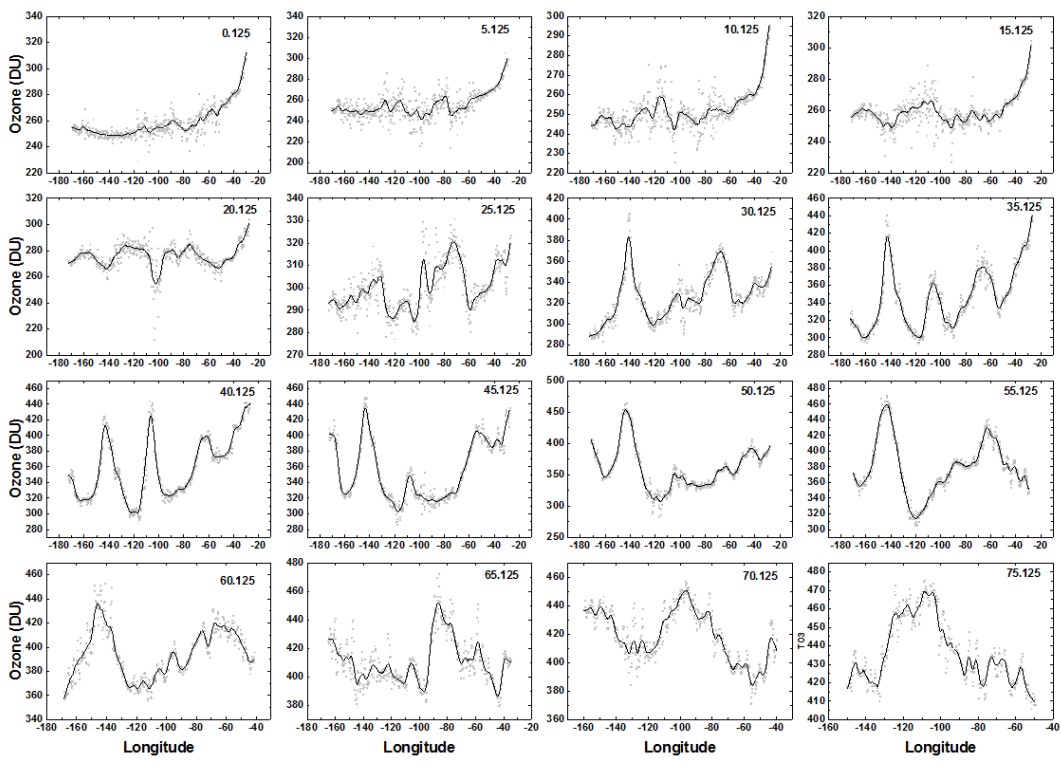

**f16**






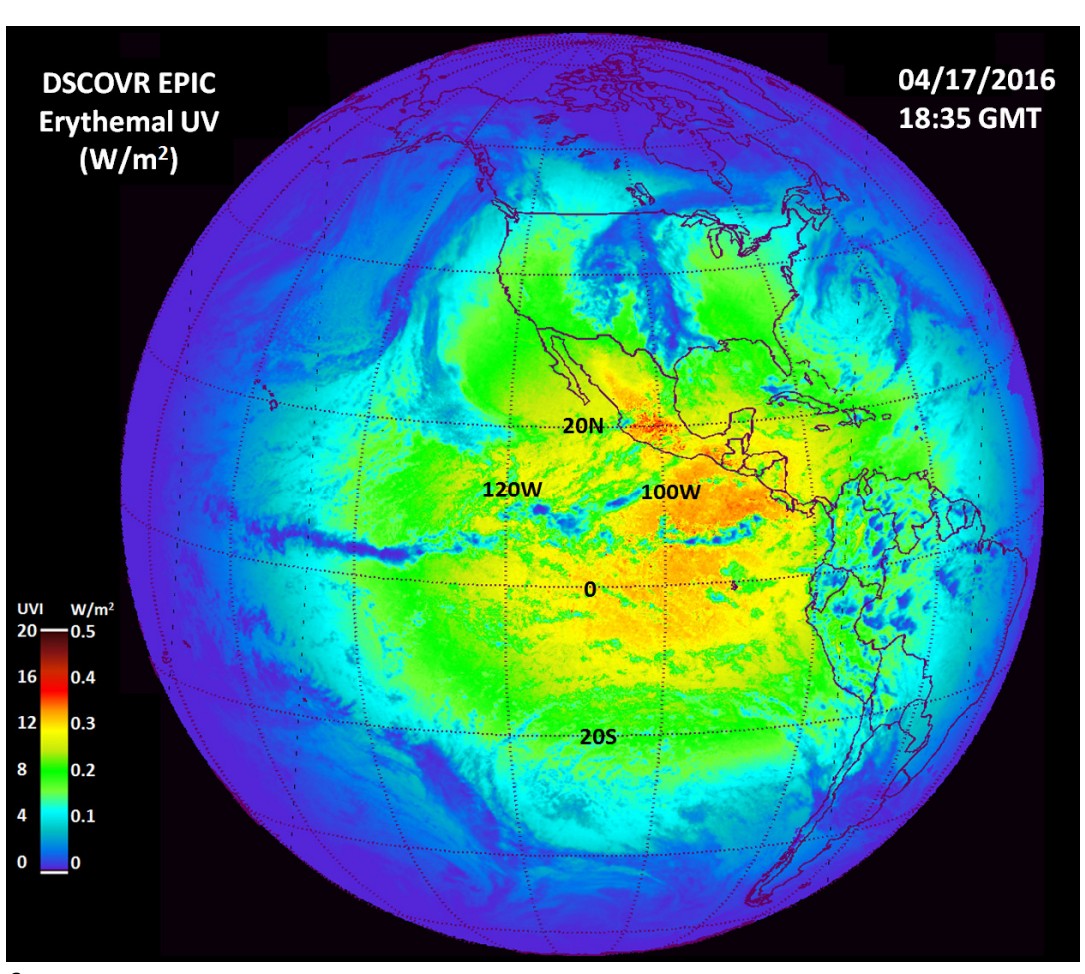

**f17**





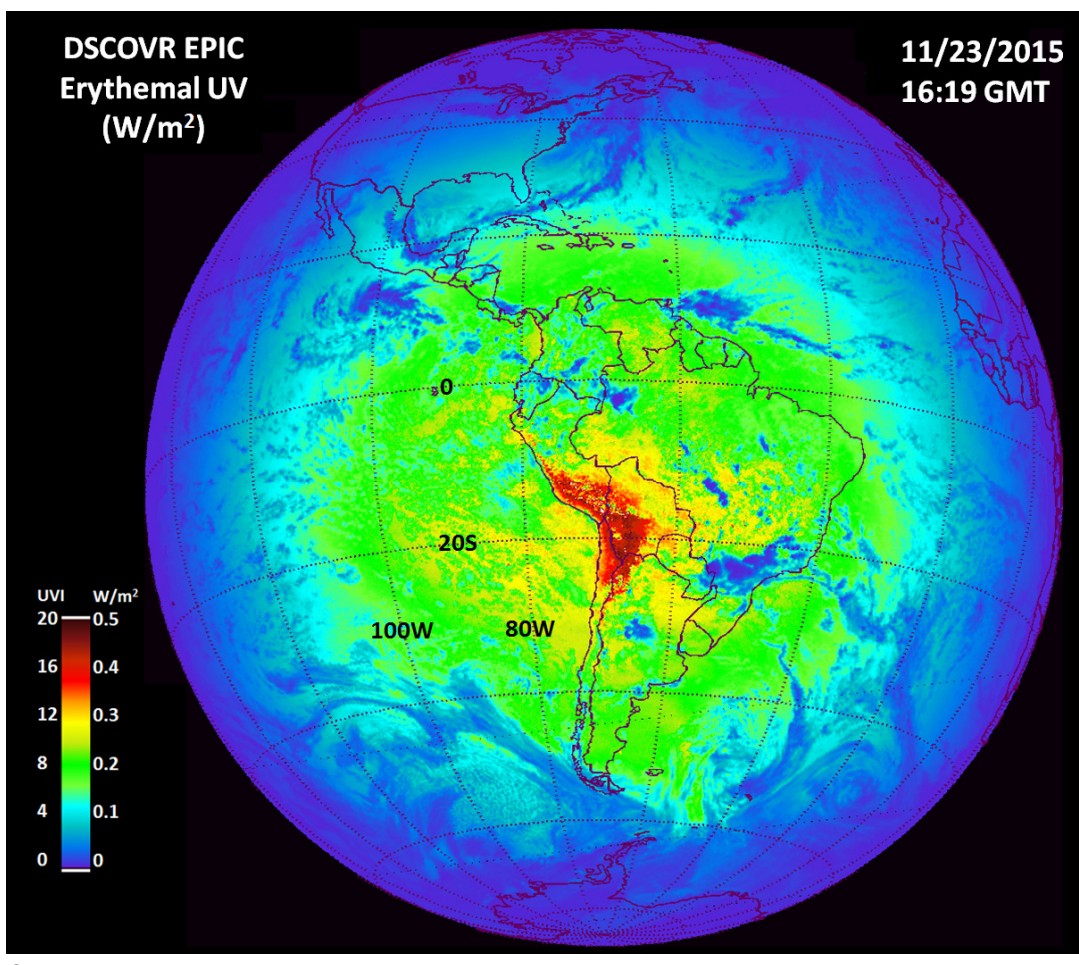

**f18**







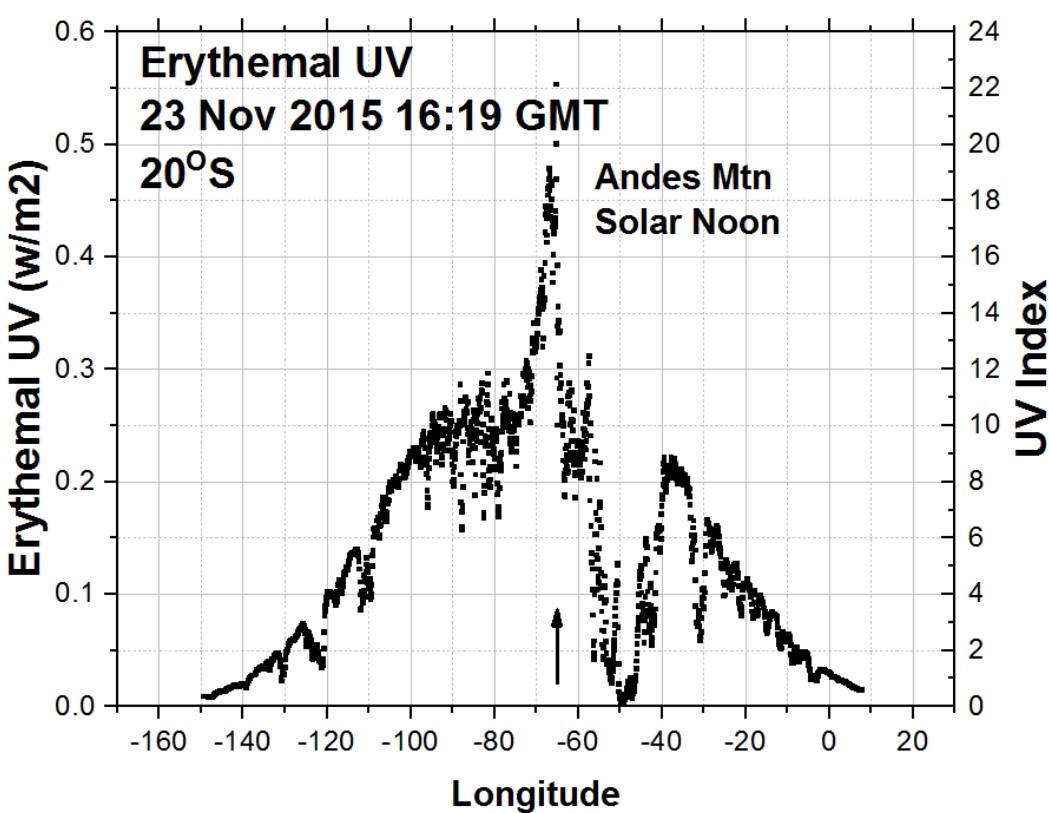

f19







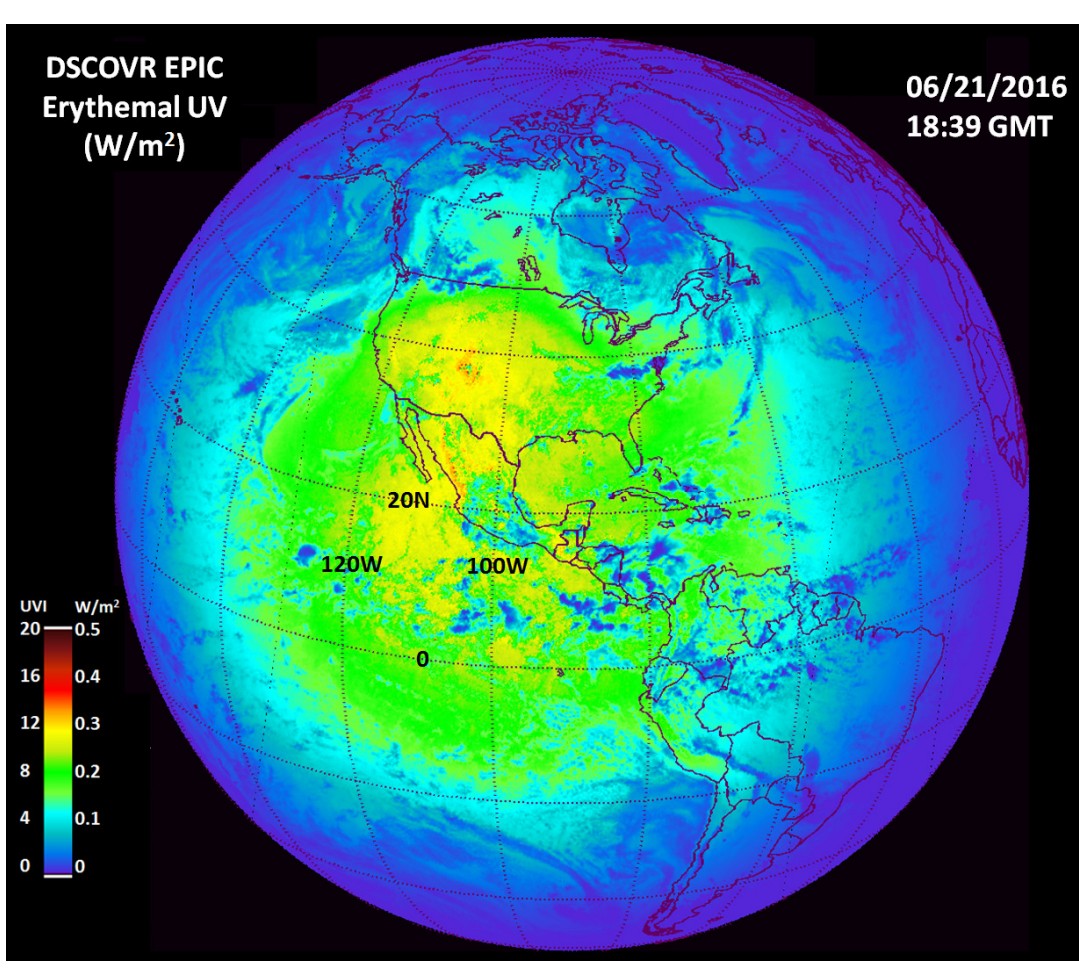

**f20**





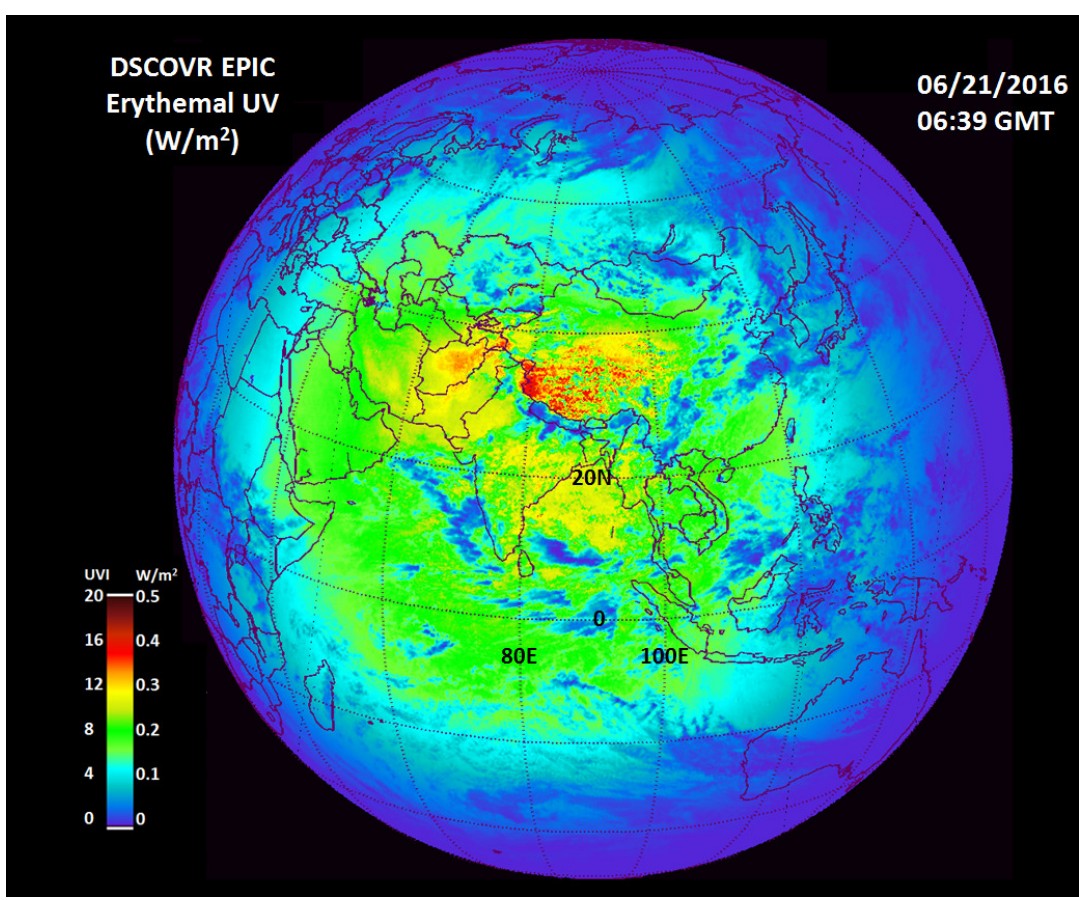

**f21**





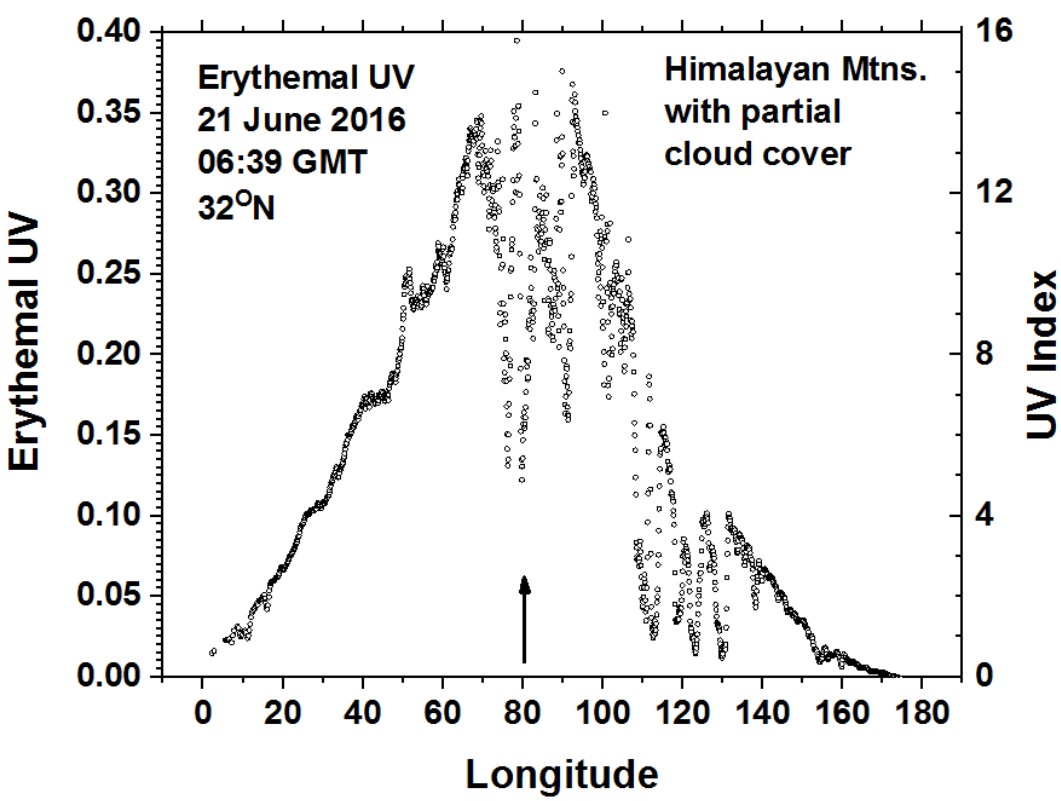

**f22**








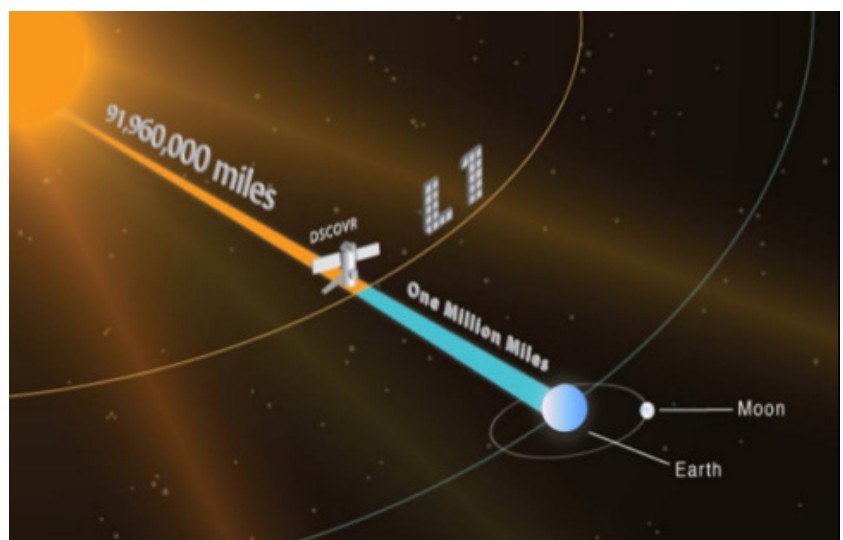

**fA1**
