# Peer review of "Synoptic Ozone, Cloud Reflectivity, and Erythemal Irradiance from Sunrise to Sunset for the Whole Earth"

_Atmospheric Measurement Techniques, 2017_

## Referee Comment (RC1) · Anonymous Referee #2 · 9 Oct 2017

The manuscript by Herman et al. describes the retrieval of ozone, cloud reflectivity, and erythemal irradiance from earth radiances measured by the EPIC instrument aboard the DSCOVR spacecraft. The paper describes in detail the retrieval of each quantity and is thus a valuable archive for end users of these products. It fits very well within the scope of AMT. General comments: The manuscript is a good resource for understanding the product retrieval from EPIC measurements. The examples of EPIC ozone and erythemal irradiance retrievals are unique and highly interesting. In my opinion the manuscript would benefit from amending the abstract and introduction with some more

context on the EPIC instrument, e.g. first of its kind? How do its products differ from and/or complement LEO satellites? Also highlighting the value of EPIC measurements for the general public (UV index – is this passed on? Published anywhere else?) would be an interesting addition.

The strictly technical tone of the manuscript makes it a bit hard to read at times. Shortening some sentences would help. Overall this paper is well suited for AMT and I recommend publication.

Specific comments: Abstract: Please include explanations of abbreviations in the abstract

p.4, line 114ff: This paragraph fits better further up in the introduction.

p.5, line 187: How can errors cancel each other out? Please explain.

p.10, line 335: Any ideas where these difference come from?

F01: air and vacuum WL in same Figure is confusing

F11: change red trace to grey/black or explain

Technical corrections:

p.1, line 9: I assume the radiances are received at the antenna in Virginia, but the derived quantities are processed elsewhere – please clarify sentence. p.2, line 33: orbit around

p.2, line 36: optimized

p.2, line 62: or over ice

p.3, line 85: 10 wavelength

p.3, line 85: at slightly

p.4, line 120: and are not

p.5, line 158: result in large

p.5, line168: correction of less

p.5, Eq. 9: for consistency, please explain omega

p.9, line 278: is shown

Section 6: several inconsistencies in use of capital "E" or not in erythemal

p.15, line 510: ensure

p.24, line 641: their

p.24, line 644: 6 months

Formatting does not fit yet AMT style

---

## Author Comment (AC2) · 19 Oct 2017

See Supplement for revised paper with changes marked in Green.

Please also note the supplement to this comment:
https://www.atmos-meas-tech-discuss.net/amt-2017-155/amt-2017-155-AC2-supplement.pdf
* * *

---

## Referee Comment (RC2) · Anonymous Referee #3 · 23 Oct 2017

General comments The objective of the paper is to provide a description of the unique Earth atmosphere observations of the EPIC camera on the DSCOVR spacecraft operating since mid-June 2015 at the Lagrange-1 point. The originality of these observations is to provide sun illuminated Earth images of the full disk every 68-110 minutes from a one million miles distance of the Earth and therefore totally different of what can be seen by a Low Earth Orbit (LEO) or Geostationary (GEO) satellites. Asides from technical details of the instrument and retrieval processes, shown in the paper are various scenes images of, LER (Lambert Equivalent Reflectivity), ozone total columns

comparisons with MEERA model and Pandora GB measurements, cloud, Erythemal irradiances, and some illustration of possible use of those data for studying longitudinal and diurnal variations of ozone and others parameters. Among those, most impressive plots are the unique Erythermal UV maps showing how useful could be those data for exploring health effect of UV radiation in the future.

Recommendation The paper is well written and easy to follow. The writing has been carefully checked. I couldn't find any remaining mistake. However, since I'm not sure that everybody knows the meaning of Lagrange-1, the only suggestion I have is to change a little the title for something like " Synoptic Ozone, Cloud reflectivity and Erythermal Irradiance for the whole Earth as viewed by the DSCOVR spacecraft from Lagrange-1 orbit at 1.5 million km from the Earth" Otherwise, since it appears to me an excellent paper which matches very well AMT editor's acceptance criteria, my recommendation is to accept Âń as it is Âż.

---

## Author Response (AR1)

I have changed the title to

EPIC Synoptic Ozone, Cloud Reflectivity, and Erythemal Irradiance from Sunrise to Sunset for the Whole Earth as viewed by the DSCOVR spacecraft from the Earth Sun Lagrange-1 Orbit

[Figure]

[Figure]

EPIC (Earth Polychromatic Imaging Camera) onboard the DSCOVR (Deep Space Climate Observatory) spacecraft is the first Earth science instrument located near the Earth-Sun gravitational plus centrifugal force balance point, Lagrange-1. EPIC measures Earth reflected radiances in 10 wavelength channels ranging from 317.5 nm to

779.5 nm. Of these channels, four are in the UV range 317.5, 325, 340, and 388 nm, which are used to retrieve O3, 388 nm scene reflectivity (LER Lambert Equivalent Reflectivity), SO2, and aerosol properties. Unlike low earth orbiting satellite instruments near noon values, these synoptic quantities for the entire sunlit globe from sunrise to sunset obtained every 68 minutes when it is summer or 110 minutes in winter at the receiving antenna in Wallops Island, Virginia. Depending on solar zenith angle, either 317.5 or 325 nm channels are combined with 340 and 388 nm to derive ozone amounts. As part of the ozone algorithm, the 388 nm channel is used to derive LER. The retrieved ozone amounts and LER are combined to derive the Erythemal irradiance for the entire sunlit Earth's surface, 2048x2048 points, at a nadir resolution of 18 x 18 km2 using a computationally efficient approximation to a radiative transfer calculation of irradiance. Corrections are made for altitude above sea level and for the reduced transmission by clouds based on retrieved LER.

Also highlighting the value of EPIC measurements for the general public (UV index – is this passed on? Published anywhere else?) would be an interesting addition . I have given the algorithm to responsible parties at NOAA. They expressed interest, but have not indicated that they are using the new approach.

Please include explanations of abbreviations in the abstract. Done p.4, line 11 ff: This paragraph fits better further up in the introduction.

I moved the paragraph into the introduction. The data and images of the changing synoptic cloud cover from sunrise to sunset are unique to the EPIC satellite instrument. Neither geostationary nor low earth orbiting satellites can produce these data or images. Geostationary satellites could produce something similar, but to date, none have the UV channels for ozone and LER, and geostationary satellites are limited to a range of approximately ±60O latitude and ±60O longitude. While low earth orbiting satellite data can be combined to produce a global representation of ozone and cloud cover, all the ozone and cloud cover are for a fixed local time (e.g., 13:30 hours for OMI) and are

not representative of the atmosphere at other times of the day. 1.1 EPIC Instrument p.5, line 187: How can errors cancel each other out? Please explain.

These are not errors. Rather to form the albedo, one takes the ratio of two quantities that both contain the solar Fraunhofer line structure. The resulting ratio does not have the Fraunhofer line structure. I modified the related sentence to read:

Because the albedo spectra AM (Eq. 1) removes the Fraunhofer line structure contained in both the solar irradiance SM and the reflected Earth radiance IM, the interpolation and convolution of AM has better accuracy than directly using IM.

p.10, line 335: Any ideas where these difference come from?

The differences arise from errors in the various satellite's calibrations.

F01: air and vacuum WL in same Figure is confusing. I chose to provide the original laboratory data (curves) done in air. I shifted the central wavelength to vacuum, since those are the values that are used in all of the science. I have added a note in the caption to clarify.

f01 Filter transmission functions (percent) for the 10 EPIC wavelengths based on laboratory measurements done in air. The central wavelength label is the shifted value used for the instrument in the vacuum of space.

F11: change red trace to grey/black or explain My error. I have fixed the figure so that all are black p.2, line 33: orbit around

I am not sure of the reviewer's reference. The text says , "to an orbit near the Earth-Sun gravitational plus centrifugal force balance point". The orbit is a Lissajous figure about the L-1 point. At times the shape of the orbit is an ellipse with the L-1 point at a focalpoint, other times the orbit is a circle with the L-1 point at the center. The orbit goes from elliptical to a circle and back to elliptical every 5 years. However, this

periodicity is altered unpredictably by in-orbit thruster corrections to account for lunar perturbations and to prevent the spacecraft from leaving the quasi-stable L-1 orbit.

The text now reads, "The DSCOVR (Deep Space Climate Observatory) spacecraft was successfully launched on 11 February 2015 to a lissajous figure orbit near the Earth-Sun gravitational plus centrifugal force balance point, Lagrange-1 (L-1), 1.5x106 km from the Earth."

p.2, line 36: optimized

The orbit was selected for earth observations, and so just turned out to be optimum for early solar storm warnings. For clarity, I changed the sentence to read, "The DSCOVR mission at L-1 is at an optimum location for early warning"

p.2, line 62: or over ice

Fresh snow over ice is correct

It turns out that the most reflective scenes are cloud-free skies with fresh snow deposited over ice. Fresh snow over land is usually not as bright because of photons lost to the absorbing ground, and scenes with only ice have a lower reflectivity because ice surfaces are usually older and darkened by pollution.

p.3, line 85: 10 wavelength Each of the 10 wavelength measurements p.3, line 85: at slightly I do not understand the reviewer's objection. The sentence reads, "Each of the 10 wavelength measurements is obtained at slightly different times.".

p.4, line 120: and are not OK (see above on page 1)

p.5, line 158: result in large OK

Section 6: several inconsistencies in use of capital "E" or not in erythemal The word "erythemal" is now all lower case except at beginnings of sentences or when combined

[revised manuscript text omitted]

longitude (2.5 hours) at 30$^{\rm O}$N–35$^{\rm O}$N midday and a longer longitudinal period 73$^{\rm O}$ (4.9 hours) in the
morning. At higher latitudes, 35$^{\rm O}$N–55$^{\rm O}$N, the variability is more pronounced with an approximate
period of 55$^{\rm O}$ (3.6 hours). In the bands from 55$^{\rm O}$N–70$^{\rm O}$N the variability is reduced and the ozone amount
falls from mid-latitude values of about 350 DU to below 300 DU. The wave structure varies throughout
the year in both hemispheres.

**5.3 Northern and Southern Hemisphere 17 April 2016 18:35 UTC**

Figure 5-5 shows the ozone retrieval for the sunlit globe on 17 April 2016 at 18:36 UTC about 1
month from the March equinox including large plumes of elevated ozone amounts (450 DU) extending
from high latitudes into mid-latitudes where the usual ozone amount is about 350 DU. For the SH (Fig. 5-
5), polar ozone variability (280-320 DU) is relatively small compared to November 23 (Fig. 10).  There is
wave structure (Fig. 15) between 30$^{\rm O}$S and 40$^{\rm O}$S with a periodicity of about 4 hours (60$^{\rm O}$ longitude) (see
also Schoeberl and Kreuger, 1983). The dip in O$_3$ amount at 77$^{\rm O}$W to 67$^{\rm O}$W and 10$^{\rm O}$S to 25$^{\rm O}$S
corresponds to the Andes Mountains in Peru, Bolivia, and Chile. While the SZA range is limited to ±70$^{\rm O}$,
the SLA reaches more than 80$^{\rm O}$ at low latitudes for longitudes between 40$^{\rm O}$S and 20$^{\rm O}$S introducing
spherical geometry correction errors that increase towards sunset near 20$^{\rm O}$W. The errors appear as
apparent increases in O$_3$ amount. At higher latitudes, the SLA is in the middle 70$^{\rm O}$s when the SZA is 70$^{\rm O}$.
The high SLA error is present in both hemispheres for observations near equinox.

The NH shows little variability in the equatorial region (0–25$^{\rm O}$N) with a mean value of about 260
DU (Fig. 16). The SLA error is present for latitudes between 0 and 15$^{\rm O}$N and 0 and 15$^{\rm O}$S that appears as
an elevated ozone amount at longitudes east of 50$^{\rm O}$W. Mid-latitudes (30$^{\rm O}$N to 40$^{\rm O}$N) show a wave
structure that is approximately 37$^{\rm O}$ apart (2.5 hours) at 35$^{\rm O}$N. A similar structure occurs in the SH with a
period of about 4.5 hours. There is an ozone maximum (red area in Fig. 14 about 450 DU) near 140$^{\rm O}$W
extending from 60$^{\rm O}$N to 35$^{\rm O}$N, very high ozone amounts in the Arctic region, and a high ozone patch
over the central US (35$^{\rm O}$N to 45$^{\rm O}$N and 104$^{\rm O}$W) peaking at 420 DU (40$^{\rm O}$N and 104$^{\rm O}$W), which probably
corresponds to a region of high atmospheric pressure.

**6.0  Estimating Erythemal Irradiance at the Earth's Surface**

The unique observing geometry of DSCOVR/EPIC permit the use of  synoptic ozone and cloud
reflectivity data to be used to compute the diurnal variation of UV irradiance from sunrise to sunset for
any point on the illuminated earth observed by EPIC.  Previous calculations from satellite data used
cloud cover and ozone from 13:30 and assumed it applied to local noon. The assumption is usually
adequate for slowly varying ozone, but not for estimating the effects of more rapidly varying cloud
cover. The following paragraphs discuss the calculation of erythemal irradiance, a spectrally weighted
mixture of UV wavelengths used as a measure of skin reddening and potential sunburn from exposure to
sunlight.

Erythemal irradiance $E_0$(SZA $\theta$, $C_T$) at the earth sea level (watts/m$^2$) is defined in terms of a
wavelength dependent weighted integral over a specified weighting function A($\lambda$) times the incident
solar irradiance I($\lambda$,$\theta$,$\Omega$,$C_T$) (Watts/m$^2$) (Eq. 11) at the Earth's sea level. The erythemal weighting function
Log$_{10}$(A$_{ERY}$($\lambda$)) is given by the standard Erythemal fitting function shown in Eq. 12 (McKinley and Diffey,
1987). Tables of radiative transfer solutions for $D_E$ = 1 AU are generated for a range of sza ($0 < \theta < 90^O$),
for ozone amounts $100 < \Omega < 600$ DU, and terrain heights $0 < Z < 5$ km using the TUV DISORT radiative
transfer model as described in Herman (2010) for erythemal and other action spectra (e.g., plant
growth, vitamin D production, cataracts, etc.).

$$E_0(\theta,\Omega,C_T) = \int_{250}^{400} I(\lambda,\theta,\Omega,C_T)A(\lambda)d\lambda \tag{11}$$

| | |
|---|---|
| $250 < \lambda < 298$ nm | Log$_{10}$(A$_{ERY}$) = 0 |
| $298 < \lambda < 328$ nm | Log$_{10}$(A$_{ERY}$) = 0.094 (298 - $\lambda$) |
| $328 < \lambda < 400$ nm | Log$_{10}$(A$_{ERY}$) = 0.015 (139 - $\lambda$) |

(12)

Equation 11 can be accurately approximated by the power law form (Eq. 13), where U($\theta$) and R($\theta$)
are fitting coefficients to the radiative transfer solutions in the form of rational fractions. Rational
fractions were chosen because they tend to behave better at the ends of the fitting range than
comparable fitting accuracy polynomials.

$$E_0(\theta,\Omega,C_T) = U(\theta) \, (\Omega/200)^{-R(\theta)} C_T \tag{13}$$

$$U(\theta) \text{ or } R(\theta) = (a+c\theta^2+ex^4)/(1+b\theta^2+d\theta^4+f\theta^6) \quad r^2 > 0.9999 \tag{14}$$

$$C_T = (1-LER)/(1-R_G) \text{ where } R_G \text{ is the reflectivity of the surface} \tag{15}$$

[revised manuscript text omitted]

f22 Erythemal Irradiances in a longitudinal slice at $32^ON$ through a portion of the Himalayan mountains.
Local solar noon is at $80.25^OE$.

fA1 An illustration of DSCOVR's Lagrange-1 orbit

**Figures**

[Figure]

**f01**

[Figure]

**f02**

[Figure]

**f03**

[Figure]

**f04**

[Figure]

**f05**

[Figure]

f06

[Figure]

**f07**

[Figure]

f08

[Figure]

**f09**

[Figure]

**f10**

[Figure]

**f11**

[Figure]

**f12**

[Figure]

**21 June 2016**
**18:41 GMT**
**25 x 25 km$^2$**

**f13**

[Figure]

**f14**

[Figure]

**f15**

[Figure]

**f16**

[Figure]

**f17**

[Figure]

**f18**

[Figure]

**f19**

[Figure]

**f20**

[Figure]

**f21**

[Figure]

**f22**

[Figure]

**fA1**

with CIE Erythemal.

p.15, line 510: ensure Careless. Now reads " EPIC's synoptic measurements ensure that"

p.24, line 641: their Now reads, "Normalized calibration functions referenced to their value at 4 Jan 2016"

p.24, line 644: 6 months Now reads, "p.24, line 644: "6 months apart"

Formatting does not fit yet AMT style. I need guidance here from the editor. I will read through the author instructions again.

―――――――――――――――――――――